# Examination of Aerosol Indirect Effects during Cirrus Cloud Evolution

Flor Vanessa Maciel[1,2], Minghui Diao[1], Ryan Patnaude[1,3]

[1]Department of Meteorology and Climate Science, San Jose State University, San Jose, 95192, USA
[2]*Current affiliation:* Department of Atmospheric and Oceanic Sciences, University of California, Los Angeles, 90095, USA
[3]*Current affiliation:* Department of Atmospheric Science, Colorado State University, Fort Collins, 80521, USA

*Correspondence to*: Minghui Diao (minghui.diao@sjsu.edu)

**Abstract.** Aerosols affect cirrus formation and evolution, yet quantification of these effects remain difficult based on in-situ observations due to the complexity of nucleation mechanisms and large variabilities in ice microphysical properties. This work employed a method to distinguish five evolution phases of cirrus clouds based on in-situ aircraft-based observations from seven U.S. National Science Foundation (NSF) and five NASA flight campaigns. Both homogeneous and heterogeneous nucleation were captured in the 1-Hz aircraft observations, inferred from the distributions of relative humidity in the nucleation phase. Using linear regressions to quantify the correlations between cirrus microphysical properties and aerosol number concentrations, we found that ice water content (IWC) and ice crystal number concentration (Ni) show strong positive correlations with larger aerosols (> 500 nm) in the nucleation phase, indicating strong contributions of heterogeneous nucleation when ice crystals first start to nucleate. For the later growth phase, IWC and Ni show similar positive correlations with larger and smaller (i.e., > 100 nm) aerosols, possibly due to fewer remaining ice nucleating particles in the later growth phase that allows more homogeneous nucleation to occur. Both 200-m and 100-km observations were compared with the nudged simulations from the National Center for Atmospheric Research (NCAR) Community Atmosphere Model version 6 (CAM6). Simulated aerosol indirect effects are weaker than the observations for both larger and smaller aerosols for in-situ cirrus, while the simulated aerosol indirect effects are closer to observations in convective cirrus. The results also indicate that simulations overestimate homogeneous freezing, underestimate heterogeneous nucleation, and underestimate the continuous formation and growth of ice crystals as cirrus clouds evolve. Observations show positive correlations of IWC, Ni and ice crystal mean diameter (Di) with respect to Na in both Northern and Southern Hemispheres (NH and SH), while the simulations show negative correlations in the SH. The observations also show higher increases of IWC and Ni in the SH under the same increase of Na than those shown in the NH, indicating higher sensitivity of cirrus microphysical properties to increases of Na in the SH than the NH. The simulations underestimate IWC by a factor of 3 – 30 in the early/later growth phase, indicating that the low bias of simulated IWC was due to insufficient continuous ice particle formation and growth. Such hypothesis is consistent with the model biases of lower frequencies of ice supersaturation and lower vertical velocity standard deviation in the early/later growth phases. Overall, these findings show that aircraft observations can capture both heterogeneous and homogeneous nucleation, and their contributions vary as cirrus clouds evolve. Future model development is also recommended

to evaluate and improve the representation of water vapor and vertical velocity on the sub-grid scale to resolve the insufficient ice particle formation and growth after the initial nucleation event.

## 1 Introduction

Cirrus clouds have varying radiative effects, depending on their microphysical properties such as ice water content (IWC), ice crystal number concentration (Ni) and mean diameter (Di) (Liou, 1992). As a result, the combined shortwave and longwave radiative effects of cirrus clouds have large spatial and temporal variabilities and may lead to either a warming or cooling effect on Earth's surface (Fu and Liou, 1993; Liou, 1986). Although cirrus clouds are ubiquitous in the atmosphere, covering approximately 20% - 40% of Earth's surface at any given time (Mace and Wrenn, 2013; Sassen et al., 2008), the processes that control their formation and evolution are spatially diverse. These processes range from the microphysical scale, such as the ice-nucleating properties of both anthropogenic and natural aerosols, to the larger dynamical scale, such as atmospheric circulations (Pruppacher and Klett, 2010). Because of the complexity of these processes, climate models have difficulties simulating them accurately and large uncertainties surrounding cirrus characteristics and the related aerosol indirect effects still exist (e.g., Boucher et al., 2013; Fan et al., 2016; Heymsfield et al., 2017; Kärcher, 2017; Lynch et al., 2002).

Cirrus clouds exist mostly within the upper troposphere and are composed of ice crystals. They can be formed through two primary mechanisms: heterogeneous nucleation and homogeneous freezing, whereas the former mechanism requires the presence of ice nucleating particles (INP) to initiate an ice nucleation event and the latter does not (Pruppacher and Klett, 2010). According to Kärcher et al. (2022), during cirrus cloud formation, a competition for available water vapor between heterogeneous nucleation on INPs and homogeneous freezing of liquid aerosols often occurs. They found that heterogeneous nucleation can deter the activation of homogeneous freezing as well as reducing the number of ice crystals formed via homogeneous freezing. Many unknowns still exist regarding the dominance and competitions between these two nucleation mechanisms. The previous work by Cziczo et al. (2013) showed that heterogeneous nucleation is likely the dominant nucleation mechanism based on in-situ observations of ice residuals sampled from cirrus in the midlatitudes. However, it is unclear if similar conclusions would be reached if other geographical locations are examined. Through the use of analytical equations, Kärcher and Jensen (2017) discovered that cirrus cloud homogeneous freezing is spatially limited and very fleeting, and ice microphysical properties are affected by both strong turbulent diffusion and entrainment mixing. For example, turbulent diffusion can dilute and spread out ice crystals formed by homogeneous freezing and expose them to ice supersaturated air for further growth, while entrainment mixing could enhance the evaporation of supercooled liquid droplets in warm cirrus regime under strong turbulence and reduce the amount of frozen droplets subsequently. Their findings indicate that a large dataset of high-resolution observations with measurements of thermodynamic and dynamical conditions would be helpful to understand the competition between different nucleation mechanisms.

Examination of the nucleation mechanisms is further complicated by the large variabilities seen in cirrus microphysical properties, associated with cirrus evolution, thermodynamic/dynamic conditions, aerosol indirect effects, and geographical

variations. Using aircraft observational data, O'Shea et al. (2016) found that in an actively growing cirrus cloud case with strong updrafts, ice crystal number concentration (Ni) was much higher compared with a decaying cirrus case by a factor of 10, and the decaying case had fewer particles larger than 700 μm. Using data from 28 flights across the arctic, midlatitudes, and tropics, Krämer et al. (2009) observed low Ni in certain cirrus clouds, which may be responsible for the elevated and persistent ice supersaturations detected inside cirrus clouds. Here, ice supersaturation (ISS) is defined as where relative humidity with respect to ice (RHi) is greater than 100%. By acting as cloud condensation nuclei (CCN) and/or ice nucleating particles (INPs), aerosols affect cloud properties and radiation (Bruce, 1989; Lohmann and Feichter, 2005; Twomey, 1977). Patnaude and Diao (2020) isolated aerosol indirect effects on cirrus clouds by restricting other conditions (e.g., thermodynamic and dynamic) and found increasing ice water content (IWC) and Ni with increasing aerosol concentrations based on multiple flight campaigns. Using satellite observations, Zhao et al. (2019) found that near cloud top in convective systems, the effects of polluted continental aerosols on ice particle effective radius vary between strong and moderate convective systems. Another study used aircraft data from the Interhemispheric Differences in Cirrus Properties from Anthropogenic Emissions (INCA) campaigns and showed that cirrus clouds in the Northern Hemisphere (NH) had higher Ni and lower effective diameters when compared with cirrus clouds in the Southern Hemisphere (SH) (Gayet et al., 2004). Based on the distributions of RHi from INCA campaigns, Haag et al. (2003) found that cirrus clouds in the SH midlatitudes are more likely to form by homogeneous freezing compared with NH midlatitude cirrus, indicating a possible role of aerosol indirect effects for controlling the in-cloud RHi distributions and the nucleation thresholds.

Several methods have been developed to compare high-resolution in-situ observations with coarser scale simulations of global climate models (GCMs). Patnaude et al. (2021) compared an in-situ observational dataset with National Center for Atmospheric Research (NCAR) Community Atmosphere Model version 6 (CAM6) simulations and found an underestimation of IWC by a factor of 3 to 10 in the NH and SH (except for SH midlatitudes), as well as weaker aerosol indirect effects in the simulations. Eidhammer et al. (2014) compared in-situ observations with CAM5 and found that simulated ice crystals have inaccurate size distributions. Using an updated CAM6 version with black carbon (BC) treated as INPs, McGraw et al. (2020) found a thinning effect followed by a thickening effect on cirrus as BC concentration increases. These studies show the importance of understanding cirrus cloud microphysical properties and aerosol indirect effects based on observations and simulations. However, a knowledge gap still exists regarding the aerosol indirect effects on different evolution phases of cirrus clouds especially during ice nucleation and growth.

In this work, we conducted an observation-based analysis using global-scale airborne in-situ measurements in order to understand the variations of cirrus microphysical properties and aerosol indirect effects at various geographical locations. In addition, the representation of these properties and processes in a GCM – NCAR CAM6 – was evaluated. We applied the method of Diao et al. (2013) to derive various evolution phases of cirrus clouds, which enables a detailed examination of aerosol indirect effects from nucleation to early growth and later growth phase. The definition of the five evolution phases is identical to the method of Diao et al. (2013). While that former study only analyzed the averaged values for each phase segment, this study mostly focuses on 1-Hz samples within each phase segment. Comparisons between observational and simulated data

were conducted through a scale-aware method, targeting cirrus microphysical properties, i.e., IWC, Ni, and mean diameter (Di), and aerosol indirect effects. In section 2, we describe the observational datasets and model simulations used for this study. In section 3, We describe the statistical distributions of IWC, Ni and Di in relation to temperature and aerosol number concentrations (Na) at various geographical locations. Section 4 discusses the implications of observed features and model biases.

## 2 Observational and Simulation Datasets

### 2.1 In-situ observations from multiple flight campaigns

For this study, we used a global-scale observational dataset comprising of seven U.S. National Science Foundation (NSF) and five NASA campaigns. The campaign name, time, location, and flight hours of the NSF campaigns are listed in Table 1. Figure 1 a and b depict the flight tracks for the NSF and NASA campaigns for the conditions of temperatures $\leq$ -40°C, respectively. We excluded measurements at temperatures > -40°C for our analysis to eliminate the existence of supercooled liquid water. These campaigns provide wide-ranging spatial coverage, spanning latitudinally from 87°N to 75°S and longitudinally from 128°E to 38°W, from the surface to the upper troposphere and lower stratosphere.

The acronyms of NSF campaigns are listed alphabetically as follows: CONTRAST (Pan et al., 2017), DC3 (Barth et al., 2015), HIPPO (Wofsy, 2011), ORCAS (Stephens et al., 2018), PREDICT (Montgomery et al., 2012), START08 (Pan et al., 2010), and TORERO (Volkamer et al., 2015). Data were collected by instruments aboard the NSF/NCAR High-Performance Instrumented Airborne Platform for Environmental Research (HIAPER) Gulfstream V (GV) research aircraft. The data are composed of 1-Hz observations of meteorological parameters such as temperature, water vapor, Na, vertical velocity (w) and cloud microphysical properties (i.e., IWC, Ni, Di). Water vapor measurements are provided by the Vertical Cavity Surface Emitting Laser (VCSEL) hygrometer aboard the aircraft (Zondlo et al., 2010). The saturation vapor pressure with respect to ice ($e_{s,ice}$) was calculated following the equation in Murphy and Koop (2005). RHi was further computed by using the water vapor mixing ratio, pressure and temperature. We used the ice crystal measurements collected by the Fast Two-Dimensional Cloud Probe (Fast-2DC) (62.5 – 3200 μm). Aerosol measurements ranging from 60 to 1000 nm were obtained by the Ultra-High Sensitivity Aerosol Spectrometer (UHSAS), which uses 99 logarithmically-spaced bins to measure their concentration and size distribution. The first deployment for HIPPO was excluded due to the absence of ice particle measurements. START08 campaign did not have aerosol measurements and was not used in the analysis of aerosol indirect effects.

Five NASA-funded aircraft campaigns are also included in this work to compare with NSF observations, including the ATTREX, MACPEX, DC3, POSIDON, and SEAC4RS campaigns (full names shown in Table 1). The main purpose of the analysis of these NASA campaigns is to provide a contrast to the NSF data from different airborne platforms and instrumentations and different types of cirrus clouds being sampled. In fact, most of the NSF campaigns (except for NSF DC3 campaign) mostly sampled in-situ cirrus clouds, while several NASA campaigns such as NASA DC3, SEAC4RS and MACPEX sampled more convective cirrus. The ice crystal measurements were collected by the 2D-S Stereo Probe (2DS) (5

– 3005 μm) for all five NASA campaigns. The water vapor measurements were collected by the Diode Laser Hygrometer (DLH) in most NASA campaigns except for the MACPEX campaign, for which the Harvard Lyman-α Photofragment Fluorescence Hygrometer was used. UHSAS instruments were used in NASA DC3 and SEAC4RS campaigns. For comparisons between NASA campaigns and CAM model simulations, two NASA campaigns – DC3 and SEAC4RS – are used

for model comparisons, since ATTREX and POSIDON did not provide aerosol measurements and MACPEX had some issues with aerosol measurements. In addition, we applied extensive quality control to this observational dataset of multiple NSF and NASA campaigns. Table S1 in the accompanying Supplement provides a list of instruments, their accuracies and precisions, measurement ranges, and related variables. Table S2 shows the UTC timestamps that were filtered out with a comment on the specific measurements found problematic. For the 1-Hz observation dataset, ice supersaturated regions (ISSR) and ice crystal

regions (ICR) were identified using values of RHi and Ni, respectively. ISSRs are regions where 1-Hz RHi is consecutively above 100%. ICRs are regions with consecutive in-cloud conditions at 1 Hz. The 1-Hz samples with at least one ice particle detected is defined as in-cloud regions. The remaining regions are defined as clear-sky conditions. The length scale of each phase segment is calculated as the number of seconds of that segment multiplying the aircraft mean true airspeed inside that segment. The average aircraft true airspeed inside all observed evolution phases is $\sim 230$ m s$^{-1}$.

We employ the method established in Diao et al. (2013), which uses the spatial ratios of ISSR and ICR to identify instances of the five evolution phases of cirrus clouds within the in-situ observations. These phases are categorized as: (1) clear-sky ISSRs, (2) nucleation, (3) early growth of ice crystals, (4) later growth of ice crystals, and (5) sedimentation/sublimation. The definitions of ISSR, ICR, and five evolution phases follow the same criteria as those described in Diao et al. (2013). The difference between this study and Diao et al. (2013) is that the previous study only analyzed the averaged conditions (such as

average RHi, IWC, Ni, and Di) for the entire evolution phase segment, while this study not only analyzes the average conditions (i.e., Figure 4) but also analyzes every second of measurements inside a phase segment by labelling each second with the phase number where that second belongs to (i.e., Figures 5 – 17). This means that in Diao et al. (2013), only one data point was used to represent one phase segment even if that segment contains many seconds of data, while this study analyzes every 1-Hz datum within each phase, which significantly increases the sample size. An example diagram illustrating the differences

between this study and Diao et al. (2013) is shown in supplemental Figure S1.

Figure 2 shows an idealized illustration of these five evolution phases. One should note that these idealized phases may appear simultaneously in the same cloud, and the time evolution may not follow the exact sequence from phase 1 to 5. These phases may also be adjacent or overlap with each other in a 3-D view, but may not be captured from 1-D sampling of in-situ airborne observations. To compare the high-resolution NSF observational data with the coarse-resolution model data, a running average

of 430 seconds was applied to the 1-Hz observation data. This timescale, i.e., 430 seconds, was chosen since it converts to a horizontal scale of about 100 km for a mean airspeed of 230 m s$^{-1}$ for all NSF campaigns and 229 m s$^{-1}$ for two NASA campaigns with a temperature less and equal to -40°C. The 430-s moving average uses one second (center point) along with 215 seconds before and 214 seconds after this second to calculate averaged values of IWC, Ni and Di, Na$_{500}$ and Na$_{100}$ on the logarithmic scale. These 430 seconds include both in-cloud and clear-sky conditions. For phase identification, cloud evolution

phase number cannot be averaged (i.e., phases 1 and 2 cannot be averaged to phase 1.5), and the evolution phase identification relies on using high-resolution observations to capture the transitioning between ICRs and ISSRs. Thus, the phase number of this center-point second is used to represent the phase number of the 430-second segment.

**2.2 CESM CAM6 model simulations**

In the NCAR Community Earth System Model 2 (CESM2) CAM6 model, a prognostic moist turbulence scheme called the
Cloud Layers Unified by Binomials (CLUBB) replaces the previous schemes for boundary layer turbulence, cloud macrophysics and shallow convection (Bogenschutz et al., 2013). An adjustment was also applied to the deep convection scheme to incorporate sensitivity to convection inhibition (Zhang and McFarlane, 1995). In addition, instead of treating hydrometeors, i.e., rain and snow, diagnostically, CAM6 includes an improved bulk two-moment cloud microphysics scheme that treats them prognostically (Gettelman and Morrison, 2015). For the simulations of aerosols and aerosol-cloud interactions
the microphysics scheme is coupled with MAM4, a four-mode aerosol model that permits ice crystals to form through the heterogeneous nucleation of dust particles and homogeneous freezing of sulfate aerosols (Liu et al., 2007; Liu and Penner, 2005). Finally, for considerations of pre-existing ice the model follows Shi et al. (2015).

  We conducted nudged simulations of the NCAR CAM6 model for each of the NSF and NASA flight campaigns. Model output collocated with their respective flight tracks was saved and used in the analysis. The specific configuration of the model
simulation is identical to that in Patnaude et al. (2021). These simulations were nudged for temperature and 2-D horizontal wind according to the Modern-Era Retrospective Analysis for Research and Applications version 2 (MERRA2) data (Gelaro et al., 2017). All nudged simulations had a spin-up time of 6-months before their respective campaign's start date. These simulations have 32 vertical levels and a horizontal resolution of 0.9 degrees by 1.25 degrees. When selecting the nearest model output to the one-dimensional flight track in vertical, we use the model vertical level with the closest temperature to the
observations.

  Similar to the observational data, the $e_{s,ice}$ is calculated using the Murphy and Koop (2005) equation, while the RHi was determined from the temperature and specific humidity. In addition, by applying the methods from Eidhammer et al. (2014), we restricted the simulated ice and snow to match the size range of either the Fast-2DC probe in NSF campaigns or the 2D-S probe in NASA campaigns. The mass and number concentrations of ice and snow were then calculated by using the integrals
of incomplete gamma functions, i.e., from 62.5 to 3200 µm for NSF campaigns, or 5 to 3005 µm for NASA campaigns. Because NSF and NASA campaigns have different size ranges for cloud hydrometeor measurements, we separately compare them with CAM6 simulations and do not combine these two observation datasets. After applying this size restriction, simulated IWC, Ni and Di were calculated by combining the ice and snow variables for the comparisons with the in-situ observations of cirrus clouds. Mass concentrations ("IWC" and "QSNOW") and number concentrations ("NUMICE" and "NSNOW") of ice
and snow are summed up, respectively.

  Three modes of simulated aerosols – Aitken, accumulation and coarse aerosol modes are individually restricted to diameters > 500 nm and > 100 nm. These size-restricted aerosol number concentrations are then summed up to represent total number

concentrations of aerosols larger than 100 nm and 500 nm (i.e., $Na_{100}$ and $Na_{500}$), respectively. The same size ranges of > 100 nm and > 500 nm are also derived from the UHSAS aerosol measurements for model comparisons. Since the coarse-scale grid box of a GCM cannot capture the high-resolution spatial relationships between ISSRs and ICRs, the method of identifying five evolution phases of cirrus clouds cannot be applied on the 1-degree grid box scale. However, since only collocated model output to the 1-D flight track is used in this analysis, the comparison between each observation datum and its nearest model point assumes that the model point has the same cirrus evolution phase as that specific observation datum. In other words, we assume that the model output has the same cloud evolution phase as the matching observations and then evaluate the associated cirrus microphysical properties and aerosol indirect effects for each phase.

## 3 Results

### 3.1 Occurrence frequencies of five cirrus evolution phases

The probabilities of ICR/ISSR spatial ratios for each campaign and various latitudinal regions are shown in Figure 3 a – c. This probability is based on cloud segment number, which is calculated as the number of cloud segments in a bin of $\log_{10}$(ICR/ISSR) spatial ratio divided by the total number of cloud segments with real values of $\log_{10}$(ICR/ISSR) in all bins, similar to the calculation in Diao et al. (2013) and their Figure 4 b. This parameter allows us to quantify how ICRs expand with respect to ISSRs. In Figure 3 a and b, the ICR/ISSR spatial ratio almost always peaks around one for each campaign, meaning that ICRs and ISSRs often have similar spatial scales in each cloud segment, which suggests that these regions are likely to coexist. The combined NSF and NASA data were further separated into six latitudinal regions: Northern Polar (NP), Northern Midlatitude (NM), Northern Tropical (NT), Southern Polar (SP), Southern Midlatitude (SM), and Southern Tropical (ST) to investigate possible hemispherical differences in cirrus cloud properties (Figure 3 c). Similar to the analysis of each campaign, ICR/ISSR spatial ratios peak at one for most of the latitudinal regions.

Five evolution phases are identified for all flight campaigns, and their probabilities are shown in Figure 3 d – f. This probability of each phase is based on cloud segment length, which is calculated as the lengths of a certain phase divided by the total lengths of all five phases. This is different from Diao et al. (2013) and their Figure 4 a, which calculated the phase probability as the number of segments in a phase divided by the total number of all segments in five phases (i.e., number-based instead of length-based). The result shows that the early growth phase (i.e., when ISSR and ICR intersect each other) has the most dominant spatial coverage for almost all the campaigns (except MACPEX). Since the ICR/ISSR ratio is also around unity, these two results indicate that the coexistence of ISSRs and ICRs at similar spatial scales provides a semi-steady state for cirrus evolution, possibly because new ice crystal formation in ISSRs can balance out the sedimentation of aged ice crystals and therefore cirrus clouds can persist in this condition. When separated by latitudinal regions, evolution phase 3 consistently shows the largest length for most regions. The previous studies by Diao et al. (2013, 2014b) showed higher probabilities of phases 1 and 5 because most of those phases have higher segment number but shorter lengths.

Occurrence frequencies of clear-sky ISS have been previously used as an indicator of ice nucleation (e.g., Ovarlez et al., 2002; Diao et al., 2014b). That is, lower clear-sky ISS frequencies indicate that ISSRs are more likely to transition into ICRs and ice nucleation is more likely to occur. This study found that NH has lower clear-sky ISSR length-based frequencies (0.16) compared with the SH (0.28) when counting the total seconds of samples. Diao et al. (2014b) found similar number-based frequencies of clear-sky ISSRs between the NH and SH (their Figure 5) when comparing the number of consecutive segments. These two findings indicate that clear-sky ISSR segments in the NH are patchier (i.e., shorter length but similar number) than those in the SH. In addition, the ORCAS campaign located around Punta Arenas, Chile, is around the same region as the southbound flights of INCA campaign. The ORCAS campaign shows the highest clear-sky ISSR frequency among all NSF and NASA campaigns (Figure 3 d), which may be the reason that this work and the previous work of Ovarlez et al. (2002) both show higher clear-sky ISS frequencies in the SH.

Examining each latitudinal region between the two hemispheres, tropical regions show similar frequencies of each evolution phase between NT and ST. For the midlatitudes, NM shows lower frequencies of clear-sky ISSR and nucleation phases, and higher frequencies of later growth and sedimentation phase compared with SM. These results indicate that higher aerosol loading in the NH midlatitude may facilitate ice crystal formation and provide a faster transition from clear-sky ISSR and nucleation phases to early/later growth and sedimentation.

Since the frequency of clear-sky ISSR or ISS is often used as an indicator of how easily ice nucleation may occur, we further compare the frequencies of clear-sky ISSRs among all regions. NM shows the lowest frequency (0.05) of clear-sky ISSRs (i.e., phase 1). For the polar regions, both NP and SP have relatively higher frequencies of clear-sky ISSRs (frequencies of 0.6 and 0.2, respectively). This is likely caused by the low temperatures at the polar regions, since a smaller magnitude of temperature cooling rate and/or vertical velocity is needed to generate the same magnitude of ice supersaturation at lower temperatures compared with higher temperatures. The higher ice supersaturation frequencies in the polar regions were also previously observed in satellite data (e.g., Gettelman et al., 2006). NP has higher frequencies of clear-sky ISSRs than the SP, possibly due to the asymmetrical sampling of more high-latitudinal regions in the NH than the SH (as shown in Figure 2 c and d).

## 3.2 Relative humidity and particle size distributions for each evolution phase

Distributions of segment-average RHi for each phase and all evolution phases are shown in Figure 4. As cirrus clouds evolve from clear-sky ISSRs to regions with both ice crystals and ice supersaturation, the RHi first increases with ISSR length for phase 1 before the first ice crystals appear, but decreases with the spatial ratio of ICR/ISSR once ice crystal formation and growth reduce the amount of available water vapor exceeding ice saturation. Subsequently, as ice particles sediment and sublimate, the decreasing ICR lengths are associated with decreasing RHi, indicating that more ice particles sediment into drier conditions as the cirrus evolves in this final phase. As a result, phase 5 also has the widest range of RHi values during sedimentation and sublimation compared with other phases. Interestingly, the highest RHi values are mostly seen in phase 2 (nucleation phase) instead of phase 1 (clear-sky ISSR), indicating that phase 2 is a better representation of the RHi threshold for ice nucleation compared with phase 1. This feature is also seen in the study of Diao et al. (2013). The maximum RHi values

seen in phases 1 and 2 from seven NSF campaigns are 173% and 174%, while those seen in NASA campaigns are 180% and 212%, respectively. Since the RHi threshold of homogeneous freezing calculated based on 0.5 μm aerosols ranges from 140% to 160%, this result indicates that homogeneous freezing has been observed and can be captured in the nucleation phase identified from the method represented here. However, since most (89%) of the nucleation phase (red markers) have RHi values less than 140% among all ice supersaturated conditions, it suggests that heterogeneous nucleation mechanism is likely the more commonly observed mechanism for ice nucleation compared with homogeneous freezing. Similar method of using the RHi distribution to indicate the dominant nucleation mechanisms of cirrus clouds from in-situ airborne observations was used by Cziczo et al. (2013). Yet this study is the first one to directly separate the nucleation phase from in-situ airborne observations and examines the RHi distribution only for the nucleation phase.

Hemispheric distributions of the five evolution phases and their frequencies of showing ISS exceeding 40% among all ISS conditions are shown in Figure 5. This analysis uses the combined dataset of NSF and NASA campaigns. Frequencies of each evolution phase are calculated by the number of samples (in seconds) of a certain phase in a specific bin divided by the total number of samples (in seconds) in that bin. The tropical regions show higher frequencies of the later growth phase (phase 4), indicating that cirrus clouds in the tropics may have prolonged lifetime with sustained availability of excess water vapor over ice saturation. The SH midlatitudes have slightly higher frequencies of clear-sky ISSRs compared with the NH midlatitudes, possibly due to higher concentrations of INPs in the NH midlatitudes. The frequencies of high RHi (> 140%) are calculated by the number of samples of RHi > 140% divided by the number of samples of RHi > 100%. Higher RHi values were observed more frequently in phase 2 in the NH and SH extratropical regions. Most bins in phase 2 show less than 0.1 frequencies for RHi exceeding 140% among all ice supersaturated conditions, indicating that heterogeneous nucleation is more dominant than homogeneous freezing in the nucleation phase.

Number concentrations of ice particles in various size bins are examined by using the particle size distribution (PSD) plots for the NSF and NASA datasets in Figure 6, separated by various campaigns, latitudes, and evolution phases. The 2DC and 2DS probes were used in NSF and NASA data, respectively. For NSF data, the DC3 campaign shows the highest particle number concentrations per bin while ORCAS has the lowest. This may be because DC3 primarily targeted anvil outflows from convective systems over the central U.S., while ORCAS sampled in-situ formed cirrus clouds over the Southern Ocean region. Convective cirrus is often seen to be associated with higher IWC and Ni (e.g., Krämer et al., 2016) compared with in-situ formed cirrus clouds. In addition, the higher Na in the continental U.S. compared with the Southern Ocean also generally leads to higher total Ni as seen in Patnaude and Diao (2020). Among all NASA campaigns, NASA DC3 has the highest particle number concentration per bin while ATTREX and POSIDON have the lowest. This may be caused by several reasons, e.g., different geographical locations (tropics versus midlatitude, land versus ocean) and different types of cirrus (i.e., DC3 sampled continental cirrus with closer proximity to convective activity). Thus, different cirrus origins and different aerosol loadings over land and ocean may both lead to different Ni in these campaigns. The aerosol indirect effects on Ni will be further discussed in Section 3.4. Latitudinal variations in PSD are examined in Figure 6 c and d. For the NSF data, the NH has higher

particle number concentrations per bin compared with its counterparts in the SH. A consistent result is seen in the NASA data with higher concentrations per bin in the NT compared with ST.

When separated by evolution phases (Figure 6 e and f), among the three phases that ice particles coexist with ice supersaturation (phases 2 – 4), the nucleation phase (phase 2) shows the lowest particle number concentration per bin for both NSF and NASA data. In addition, the later growth phase (phase 4) shows the highest particle number concentration per bin when the size is

300 less than 1700 μm, likely due to continuous ice crystal formation and growth with a sufficient supply of water vapor. The previous study of Diao et al. (2013) also observed an increasing Ni along the time evolution of cirrus clouds, while Spichtinger and Gierens (2009) showed similar increasing Ni trend along cirrus evolution using a box model.

### 3.3 Comparisons of cirrus microphysical properties between observations and simulations

The relationships of $\log_{10}$(IWC), $\log_{10}$(Ni) and Di with respect to temperature are shown for each of the in-cloud evolution

phases (2 – 5) for 1-s observations, 430-s averaged observations from NSF campaigns and CAM6 simulations in Figure 7. Because the NSF and NASA campaigns have different size ranges for ice crystal measurements as described in Section 2.2, the following analysis in Sections 3.3 – 3.6 only shows six NSF campaigns (since START08 did not have aerosol measurements), except for Figures 12 and 13 which show two NASA campaigns.

For both higher and lower resolution observations, phase 4 shows the higher IWC and Ni, followed by phase 3, phase 5 and

310 phase 2, demonstrating that IWC and Ni continue to increase from nucleation phase to later growth phase but decrease as the evolution proceeds to sedimentation/sublimation. This trend is consistent with the analysis of PSD (Figure 6) as well as the previous study of Diao et al. (2013). CAM6 simulations are able to capture the same evolution trend of IWC and Ni as the observations. The model also captures the increasing trend of IWC, Ni and Di with increasing temperature as seen in the observations.

One of the main differences between the simulations and observations is the much lower simulated IWC in phases 3 and 4 (early and later growth phases) by 0.5 and 1–1.5 orders of magnitude, respectively. In addition, the simulated Ni in phases 2 and 3 (nucleation and early growth) is much higher than the observed values by 1 and 0.5 orders of magnitude, respectively. Simulated Di is underestimated compared with observations by a factor of 1.5 – 2 for all phases, especially at higher temperatures. The lower IWC, higher Ni and lower Di in relation to temperature seen in the model have also been shown in

Patnaude et al. (2021). In addition, the model shows smaller variations of IWC and Ni between the nucleation phase and later growth phase compared with the observations. For both observations and simulations, Di does not show significant variations among different evolution phases. These results indicate that simulations may overestimate the contribution of homogeneous freezing in the nucleation phase (i.e., represented by too high Ni and too low Di), as well as underestimating growth of ice particles after ice nucleation.

We also compared CAM6 simulations with two NASA campaigns (shown in Figure S2) and found similar main features as Figure 7, that is, the simulations show smaller variations between different cirrus evolution phases, higher Ni in phase 2 but lower Ni in phases 3 and 4, as well as lower IWC in later phases.

## 3.4 Aerosol indirect effects and nucleation mechanisms during cirrus evolution

The relationships between aerosol number concentrations and cirrus microphysical properties are examined for smaller and larger aerosols (Figures 8 and 9, respectively). Both 1-s and 430-s observations show increasing IWC and $N_i$ with increasing $Na_{100}$, indicating aerosol indirect effects of smaller aerosols (> 100 nm) for facilitating ice crystal formation. Compared with $Na_{100}$, $Na_{500}$ shows the most significant positive correlations with IWC and $N_i$ for the nucleation phase especially at $Na_{500} > 2$ $cm^{-3}$ (i.e., $log_{10}Na_{500} > 0.3$), followed by the early growth phase, then later growth phase. This result indicates that large aerosols (> 500 nm) likely act as INPs to initiate ice nucleation, and heterogeneous nucleation occurs frequently during the nucleation phase in the observations. Compared with the observations, simulations do not show a significant trend of cirrus microphysical properties in relation to either $Na_{100}$ or $Na_{500}$.

Similar to Figure 7, both 1-s and 430-s observations show a similar trend of increasing IWC and $N_i$ as cirrus evolves from phase 2, 3 to 4. An interesting difference between $Na_{100}$ and $Na_{500}$ is seen for their impacts on phases 2 – 4. For the mean diameter $D_i$, a slight increase of $D_i$ is seen with increasing $Na_{100}$ and $Na_{500}$. The increase of $D_i$ is more significant with increasing $Na_{500}$ than $Na_{100}$, especially for phases 2 and 3. The average $D_i$ for five phases is mostly at or below 200 μm when examined against various $Na_{100}$ values. In contrast to that, the average $D_i$ exceeds 200 μm when $Na_{500}$ exceeds 10 $cm^{-3}$ for phases 2–4. The phases 3 and 4 even reach $D_i$ around 450 μm at higher $Na_{500}$. This further indicates that $Na_{100}$ are more likely correlated with homogeneous freezing, which produces smaller ice crystals, while $Na_{500}$ are more correlated with heterogeneous nucleation, which produces larger ice crystals.

Differing from the observations, the simulated IWC, $N_i$ and $D_i$ are almost identical among various cirrus evolution phases at various $Na_{100}$ bins, while the observations show increasing IWC and $N_i$ as well as decreasing $D_i$ as cirrus evolves. Differing from Figure 7 which shows that model overestimates $N_i$ at various temperatures, Figures 8 and 9 show that model initially overestimates $N_i$ for nucleation phase, but then underestimates $N_i$ for later growth phase, indicating that the model initially overestimates homogeneous freezing but later has insufficient formation of new ice crystals as cirrus evolves.

The missing variations among different evolution phases indicate several potential biases in the model – 1) a lack of representation of the evolving role of heterogeneous and homogeneous nucleation (in contrast to the observations, which show higher $D_i$ initially from heterogeneous nucleation followed by lower $D_i$ from both heterogeneous and homogeneous nucleation); 2) insufficient growth of ice particles after ice nucleation indicated by lower IWC (in contrast to observed increasing trend of IWC as cirrus evolves); 3) insufficient new ice crystal formation in early and later growth phases (i.e., in contrast to observed increasing trend of $N_i$ as cirrus evolves). We further discuss the first factor in the rest of this Section 3.4 and discuss the second and third factors through diagnosis of thermodynamic and dynamic conditions in Section 3.6.

To quantify aerosol indirect effects on cirrus microphysical properties, linear regressions were applied to the 1-s observations, 430-s observations and model simulations (Figures 10 and 11). Differing from the analyses in Figures 8 and 9, delta values were calculated for logarithmic IWC, $N_i$, $D_i$, $Na_{100}$ and $Na_{500}$. That is, for each variable, we calculated the mean value of that variable in each 1-degree temperature bin, and then subtracted these mean values from each datum (at either 1-s or 430-s

resolution) based on the temperature bin that the datum belongs to. In other words, these delta values remove the general trend of each variable with changing temperature (as shown in Figure 7). Patnaude and Diao (2020) showed that restricting the temperature influence before quantifying aerosol indirect effects is very important as the temperature can be a major factor affecting cirrus microphysical properties. Thus, delta values are also used in this study to minimize the impact of temperature

when analyzing aerosol indirect effects. Table S3 in the Supplement documents the full linear regression equations, including slopes, intercepts, and their standard deviations. Indicators of statistical significance such as $R^2$ values and p-values are also shown in that table. After restricting the temperature influences, positive correlations are seen for IWC and $N_i$ with respect to both $Na_{100}$ and $Na_{500}$ for all phases with ice supersaturation (i.e., phases 2 – 4) in the 1-s observations. For the 430-s observations, positive correlations are also seen for phases 3 and 4, but the correlations become insignificant for phase 2. This

is due to the spatial averaging process that includes clear-sky segments as part of the grid averages. Since phase 2 (nucleation phase) generally has shorter lengths of ICRs compared with phases 3 and 4 (as shown in Figure 4), the averaging process leads to lower IWC and $N_i$ for phase 2 compared with phases 3 and 4. Due to this reason and the fewer samples of phase 2, the distributions of IWC and $N_i$ also show more fluctuations in the coarser scale observations.

Aerosol indirect effects on IWC and $N_i$ are quantified by the slope values of linear regressions (given in the figure legend).

The slope values between IWC and $N_i$ are comparable with each other for the observations on the same scale (e.g., Figure 10 a and d, b and e), but show lower values for 430-s observations compared with 1-s observations. Nevertheless, the positive slope values are consistently seen for phases 3 and 4 between 1-s and 430-s observations against both $Na_{100}$ and $Na_{500}$.

Compared with observations, the simulations show either negative correlation or no significant correlation with respect to $Na_{100}$. As for the impact of $Na_{500}$, the simulations show a slight positive correlation between $N_i$ and $Na_{500}$ in phase 4 (later

growth phase), but a negative correlation is seen for phase 2 (nucleation phase). This result indicates that the model may have weaker aerosol indirect effects from larger aerosols to activate ice nucleation than those seen in the observations.

Contrasting the role of smaller and larger aerosols based on high-resolution observations, the smaller aerosols show the strongest positive correlations with IWC, $N_i$ and $D_i$ in phase 4. On the contrary, the larger aerosols show the strongest correlations with these properties in phase 2. This feature suggests that when ice nucleation just starts to occur, the larger

aerosols which often include INPs likely dominate ice nucleation in the nucleation phase. The dominance of heterogeneous freezing on the nucleation phase can be seen from the slope values of IWC and $N_i$, i.e., 1.2 and 0.98 ($R^2 = 0.61$ and 0.70) with respect to $Na_{500}$, which are about 3.6 – 3.8 times of the slope values of 0.33 and 0.26 ($R^2 = 0.19$ and 0.27) with respect to $Na_{100}$, respectively. The higher $R^2$ values for $Na_{500}$ also indicate higher statistical significance for the correlations with $Na_{500}$. In addition, the nucleation phase shows higher slope value of $D_i$ with respect to $Na_{500}$ (i.e., 0.07), while the slope is negative

(-0.003) with respect to $Na_{100}$. In addition, IWC and $N_i$ slowly increase with increasing $dlog_{10}Na_{100}$ when it is relatively low (i.e., $dlog_{10}Na_{100} < 1$), yet they significantly increase when $Na_{100}$ is 10 $cm^{-3}$ higher than the mean $Na_{100}$ value. On the other hand, IWC and $N_i$ continuously increase with $dlog_{10}Na_{500}$ throughout the entire range. This feature indicates that at lower $Na_{100}$, aerosols may activate ice nucleation through homogeneous freezing such as by freezing sulfate aerosols, while at higher

$Na_{100}$, there is a higher probability of having INPs that can initiate ice nucleation through heterogeneous nucleation and further expedite the formation of new ice crystals.

As cirrus evolves with additional ice supersaturation, more ice nucleation events start to take place possibly with similar amounts of contributions from homogeneous freezing and heterogeneous nucleation. The comparisons of two nucleation mechanisms are illustrated in the slope values of IWC and Ni for phase 4, that is, 0.43 and 0.16 with respect to $Na_{500}$, similar to the slope values of 0.41 and 0.23 with respect to $Na_{100}$, respectively. This feature is likely caused by a gradual depletion of INPs in the previous nucleation events through heterogeneous nucleation unless a continuous supply of new INPs is available, resulting in comparable contributions from both nucleation mechanisms. Note that having contributions from homogeneous freezing and heterogeneous nucleation inside a cirrus cloud does not necessarily mean that these two nucleation mechanisms compete for water vapor, since they may occur at different locations and times inside a cloud. The idea that multiple nucleation events may continuously occur in the lifetime of a cirrus cloud is also supported by the existence of ISSRs either overlapping with or adjacent to ICRs. Thus, the observations of higher Ni in early/later growth phases do not contradict with box model simulations, which usually show a lower Ni when homogeneous freezing directly competes with heterogeneous nucleation under a fixed amount of excess water vapor over saturation.

In Figures 12 and 13, similar directions of aerosol indirect effects on IWC, Ni and Di are seen in two NASA campaigns but with higher slope values, indicating slightly stronger aerosol indirect effects in convective cirrus than in-situ cirrus. The detailed information of linear fittings and their statistical significance is shown in Table S4. Interestingly, CAM6 simulations for NASA DC3 and SEAC4RS campaigns show positive correlations of IWC and Ni with respect to Na (both $Na_{100}$ and $Na_{500}$), and negative correlation of Di with respect to Na. The simulations show stronger aerosol indirect effects in nucleation phase than early/later growth phase, as well as similar indirect effects between small and large aerosols. To further investigate the reason why model shows better agreement with observations for aerosol indirect effects on IWC and Ni during two NASA campaigns, we individually quantify aerosol indirect effects on cirrus clouds for each campaign, including six NSF and two NASA campaigns, as shown in Figures S3 and S4 in the supplement. The results show that simulations seem to better capture aerosol indirect effects for those flight campaigns targeting continental convective activity (e.g., NSF DC3, NASA DC3, and NASA SEAC4RS). This suggests that CAM6 simulations may represent the homogeneous freezing (which occurs more frequently with strong updrafts and fast cooling rate in convective cirrus) better than heterogeneous nucleation (which requires existence of INPs). This speculation is also consistent with the finding that simulations overestimate Ni and underestimate Di in nucleation phase as shown in Figures 8. Simulations show negative correlations between Di and Na while observations show positive correlations. This feature is mostly likely caused by the fact that simulations have limited amount of ice supersaturation for one-time nucleation events, and therefore ice crystals often compete for their growth (i.e., higher Ni associated with lower Di). But as we discussed above, observations may not have direct competition between multiple nucleation events if they do not happen at the same location and time as a cloud evolves. Further discussions on thermodynamic and dynamic conditions in the simulations are included in Section 3.6.

In addition, to investigate whether $Na_{500}$ and $Na_{100}$ in the model have very different values compared with the observations, distributions of $Na_{500}$ and $Na_{100}$ with respect to temperature are shown in Figures S5 and S6 in the Supplement for comparisons with NSF and NASA flight campaigns, respectively. Compared with 100-km resolution observations, the model shows similar $Na_{100}$ (within 0.5 order of magnitude) and higher $Na_{500}$ by $0.5 - 1.5$ orders of magnitude. This suggests the weaker aerosol indirect effects in the model are less likely caused by lower Na in the simulations.

**3.5 Latitudinal variations of aerosol indirect effects on cirrus clouds**

A further investigation on the variations of cirrus microphysical properties during their evolution is conducted for different latitudinal regions: the tropics, midlatitudes and polar regions in the NH and SH (Figure 14). Number of samples used for the hemispheric comparisons in Figures 14 and 15 is shown in Figures S2 and S3 in the Supplement. The geometric means of IWC, Ni and linear averages of Di at various temperatures are shown for phases 2+3+4 (i.e., phases with ice supersaturation, top three rows) and sedimenting phase 5 (bottom three rows). The largest hemispheric differences in IWC and Ni for all phases $2 - 5$ are seen in the midlatitudes, with about 1–2 orders of magnitude of higher IWC and Ni in the NH than SH based on 1-s and 430-s observations. The simulations also capture the hemispheric differences in the midlatitudes, but only show 0.5 order of magnitude of difference. Different from the midlatitudes, the polar regions show higher Di in the SH, while the tropics do not show significant hemispheric differences.

The aerosol indirect effects are contrasted between the NH and SH by combining phases $2 - 4$ (Figure 15). Table S5 shows detailed information of linear fittings for this figure. Comparing the two hemispheres, the 1-Hz observations show stronger aerosol indirect effects on IWC and Ni in the SH (i.e., larger slope values for the positive correlations) compared with the NH. The stronger aerosol indirect effects are seen from both smaller and larger aerosols, indicating that increasing the same amount of aerosol concentrations in the SH may be more effective in increasing ice nucleation compared with the NH. This suggests that even though on average the SH has lower INP concentrations than the NH, adding the same amount of INPs in the SH can have more significant impacts on cirrus microphysical properties compared with the NH. This feature also indicates that ice crystal formation in the SH may be more restricted by INP concentrations, while the NH may be more restricted by the availability of ice supersaturation. Similar to Figures 10 and 11, model simulations show weaker aerosol indirect effects compared with observations for the NH, and the model even shows negative correlations for IWC and Ni with respect to $Na_{100}$ and $Na_{500}$ in the SH. Whether this model bias is caused by the inaccurate representations of INPs or aerosol indirect effects in the two hemispheres needs more investigation in future studies.

**3.6 Diagnosis of model biases due to thermodynamic and dynamic conditions**

The model limitations in the representations of cirrus microphysical properties at various evolution phases are assessed by their thermodynamic and dynamic conditions in Figures 16 and 17, respectively. Figure 16 examines the distributions of RHi in relation to temperature for 1-s observations, 430-s observations, and model simulations. Comparing the two scales of observations, the coarser scale observations show lower magnitude of ice supersaturation for phases 1–4, due to the spatial

averaging that smooths the fluctuations of RHi. The simulations show lower frequency of ice supersaturation compared with the 430-s observations, even though the maximum RHi is similar between the simulations and the 430-s observations. That is, the model simulations show 28%, 25%, 31%, and 31% of the total samples above ice saturation for phases 1 – 4, respectively. The 1-s observations show 100%, 100%, 63%, and 45%, respectively. The 430-s observations show 49%, 50%, 45%, and 27%, respectively. The lower frequency of ice supersaturation likely leads to the lower IWC seen in the early and later growth phases (Figure 7).

Figure 17 shows the distribution of vertical velocity (w) in relation to RHi for various datasets. To better compare with model results, $\sigma_w$, or standard deviation of w, is calculated for every 40 seconds or 430 seconds of observations at 1-Hz resolution. The $\sigma_w$ values from the observations are compared with the model output variable $w_{sub}$, similar to the previous method by Patnaude et al. (2021). The 1-s observations show higher $\sigma_w$ values compared with 430-s observations, and the highest $\sigma_w$ values are seen around 4 m/s and 3 m/s for 1-s and 430-s observations, respectively. The model simulations show maximum $\sigma_w$ values around 1 m/s, much lower than the observations. 90% of the 1-Hz observations, 430-s observations, and simulation data show $\sigma_w$ values lower than 0.54, 0.63, 0.052 m/s, respectively. Low biases in the simulated $\sigma_w$ values were also seen in Patnaude et al. (2021). Simulations compared with NASA campaigns also show smaller amounts of ice supersaturation and smaller variabilities of vertical velocity (Figures S9 and S10 in the supplement).

According to Gettelman et al. (2010), $w_{sub}$ represents sub-grid vertical velocity for ice nucleation, which equals to the square root of two thirds of the Turbulent Kinetic Energy (TKE). Here TKE is defined in Bretherton and Park (2009). This calculation of $w_{sub}$ in CAM6 follows the parameterization used in Morrison and Pinto (2005). The variability of vertical velocity at sub-grid scale in the CAM6 model affects ice microphysical properties through both homogeneous and heterogeneous nucleation (Liu et al., 2007). For instance, the threshold RH for homogeneous ice formation in the slow growth regime is parameterized as a function of freezing temperature and w, while the $N_i$ of the fast growth regime is parameterized as a function of temperature and w. Therefore, both the lower $\sigma_w$ values and lower ice supersaturation frequencies can contribute to the lower IWC and lower $N_i$ in the model, especially for early and later growth phases, when significant amount of ice supersaturation and high $\sigma_w$ values ($> 1$ m s$^{-1}$) are seen in the observations but not shown in the simulations.

**4 Discussion and Conclusions**

Cirrus clouds affect radiation budget significantly with their ubiquitous coverage and vertical locations at high altitudes. In a changing climate, it is imperative to improve the understanding of cirrus cloud formation, evolution, and their relationship with aerosols. This study provides a first look of aerosol indirect effects on various evolution phases of cirrus clouds at temperatures $\leq$ -40°C based on the method of Diao et al. (2013), by using multiple flight campaigns and global climate model simulations.

The contributions of heterogeneous and homogeneous nucleation have been inferred from two types of analyses, including RHi distributions with respect to temperature (Figures 4, 5 and 16) and aerosol indirect effects using linear regressions (Figures

10 – 13). Both analyses have been separated into five evolution phases, with a specific target on phases 2 – 4 when ice supersaturation is available, that is, nucleation, early growth, and later growth phases. The RHi distributions for phase 2 show the highest magnitude (~160% to 180%) compared with phase 1 (clear-sky ice supersaturation), indicating that in-situ, high-resolution (1-Hz) airborne observations are capable of capturing homogeneous freezing, even though this mechanism was shown to be transient and small-scale based on model simulations (Kärcher and Jensen, 2017). Using the slope values to quantify correlations of IWC and $N_i$ with respect to $N_a$ (either $N_{a100}$ or $N_{a500}$), the nucleation phase shows strong positive correlations with $N_{a500}$, indicating a dominant contribution from heterogeneous nucleation. Comparatively, phase 4 shows positive correlations of IWC and $N_i$ in relation to both $N_{a500}$ and $N_{a100}$ with similar slope values, indicating similar contributions from two nucleation mechanisms for the later growth phase. This evolving weight of contribution from two nucleation mechanisms indicates that one nucleation mechanism may not dominate over the entire lifetime of a cirrus cloud, but rather part of its evolution.

Several main model limitations have been identified, including lower IWC and lower $D_i$ for all phases, higher $N_i$ for nucleation phase and lower $N_i$ for later growth phase, and lack of variations in ice microphysical properties as cirrus clouds evolve. For the underestimation of IWC and $D_i$ in the model, Patnaude et al. (2021) has previously identified part of this issue but did not provide an investigation on the possible causes. In this work, after separating five evolution phases, we categorize the potential problems in the model simulations of cirrus clouds into three main areas, including (1) overestimation of homogeneous freezing and underestimation of heterogeneous nucleation in the nucleation phase, (2) insufficient ice crystal formation and growth in early and later growth phases, and (3) weaker aerosol indirect effects for in-situ cirrus. First, the possible overestimation of homogeneous freezing in the CAM6 model is indicated by the overestimation of $N_i$ and underestimation of $D_i$ in phase 2 (Figures 8 and 9). Such underestimation of $D_i$ is most severe for phase 2 compared with later phases. In fact, both NSF and NASA observations show larger aerosol indirect effects of larger aerosols in phase 2 compared with later phases, indicating that heterogeneous nucleation plays a more significant role early on (Figures 11 and 13). Second, as cirrus evolves, simulated $N_i$ biases change from overestimation in phase 2 to underestimation in phases 3 and 4, and simulated IWC becomes more severely underestimated in phases 3 and 4, indicating insufficient ice crystal formation and growth. This feature is corroborated by the lower frequency of ice supersaturation and lower magnitudes of vertical velocity fluctuations seen in phases 3 and 4 in the simulations (Figures 16 and 17, respectively). Third, the weaker aerosol indirect effects seen in simulations for in-situ cirrus compared with convective cirrus also suggests that the model more proficiently represents homogeneous freezing than heterogeneous nucleation, since the former mechanism happens more frequently around convective activity with higher ice supersaturation. The issue of weaker aerosol indirect effects for in-situ cirrus in the simulations could be caused by either an insufficient amount of INPs considered in the model, or by the lower local ice supersaturation within each grid box at sub-grid scale. Previously, Diao et al. (2014a) showed that ice supersaturation occurs frequently at the scale of hundreds of meters, which is much lower than the GCM grid scale. Thus, future model development is recommended to investigate the representation of sub-grid scale RHi and $w$ in order to allow sufficient thermodynamic and dynamic conditions for ice

nucleation. In addition, more investigation is warranted to examine the types of aerosols that could be potentially lacking in the GCM as INPs.

Comparisons among various latitudinal regions have been conducted for tropics, midlatitudes and polar regions. The simulations show smaller hemispheric differences in the midlatitudinal regions compared with the observations, possibly due to weaker aerosol indirect effects. Comparing the two hemispheres, larger and smaller aerosols both show stronger aerosol indirect effects in the SH than the NH, indicated by the higher sensitivity of ice microphysical properties to the same amount of increase of Na in the SH compared with the NH. This result suggests that ice nucleation in the NH and SH may be limited by thermodynamic/dynamic conditions (e.g., amounts of ice supersaturation) and INP concentrations, respectively, which is also corroborated by the lower probability of clear-sky ISSRs in the NH midlatitudes than the SH midlatitudes shown in Figure 3. These differences in aerosol indirect effects reflect the different hemispheric distributions of INPs as well as other conditions (thermodynamic and dynamic). Limited by the availability of aerosol composition measurements in these flight campaigns at cirrus altitudes, future studies are recommended to investigate the aerosol indirect effects associated with different aerosol compositions at various geographical locations.

Limitations associated with the cirrus cloud evolution phase identification need to be cautioned. Ice crystals that are generated near ice saturation may be mis-represented between different phases because this method requires the usage of RHi for distinguishing evolution phases, and the RHi measurement uncertainties are around 6% – 7%. In addition, because aircraft in-situ observations only provide 1-D sampling, information about the possible cloud layers above or below the flight track is not available. For instance, ice crystals that fall from higher altitudes into lower-altitude ISSRs may potentially be treated incorrectly as newly formed ice crystals.

Overall, these results show the importance of considering the evolutionary phase of a cirrus cloud when analyzing their microphysical properties and the indirect effects of aerosols. Without separating out the phases, these cloud characteristics represent a mixture of all evolution phases, which have large variations and different responses to aerosols. When analyzing aerosol indirect effects, phase 5 is not recommended to be included as part of the analysis due to the lack of ice supersaturation. This work also demonstrated the application of a global-scale observation datasets by combining multiple flight campaigns with detailed quality control. The variations in geographical locations for cirrus microphysical properties and aerosol indirect effects can be used for evaluation of other GCM simulations. Identification of the discrepancies in cirrus characteristics and the diagnosis of the possible underlying reasons (effects of aerosols, thermodynamic and dynamic conditions) provide a pathway to better parameterize cirrus clouds and the factors influencing them at various spatial scales.

**Code availability**

The CAM6 model nudged simulations for the NSF and NASA campaigns used in this work are stored publicly on data.mendeley.com using doi:10.17632/99hdjty6sb.1 and doi:10.17632/fjw3zw2p6g.1, respectively.

## Data availability

Observations from the seven NSF flight campaigns are accessible at https://data.eol.ucar.edu/. Observations from the five NASA flight campaigns are accessible at https://www-air.larc.nasa.gov/missions.html.

## Author contributions

F. Maciel and M. Diao contributed to the development of the ideas, conducted quality control to aircraft-based observations, and wrote the manuscript. F. Maciel contributed to the subsequent data analysis. R. Patnaude generated all the model simulations.

## Competing interests

The authors declare that they have no conflict of interest.

## Acknowledgments

F. Maciel, M. Diao and R. Patnaude acknowledge funding from NSF AGS #1642291 and NSF OPP #1744965. M. Diao also acknowledges funding from NASA ROSES-2020 80NSSC21K1457. F. Maciel also acknowledges support from the San Jose State University Walker Fellowship.

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

**Table 1.** A list of the funding agency, field campaign name, date, and flight hours at temperatures ≤ -40°C for the seven NSF research campaigns and five NASA campaigns used in this work.

| Agency | Field Campaign | Full Name | Year(s) | Spatial Extent | T ≤ -40°C Flight Hours |
|---|---|---|---|---|---|
| NSF | CONTRAST | CONvective TRansport of Active Species in the Tropics | 2014 | 20°S – 40°N, 132°E – 105°W | 71 |
| | DC3 | Deep Convective Clouds and Chemistry Project | 2012 | 25°N – 43°N, 106°W – 79°W | 73 |
| | HIPPO* | HIAPER Pole-to-pole Observations | 2009 2010 2011 | 67°N – 87°N, 128°E – 90°W | 118 |
| | ORCAS | The $O_2/N_2$ Ratio and $CO_2$ Airborne Southern Ocean Study | 2016 | 75°S – 18°S, 91°W – 51°W | 41 |
| | PREDICT | PRE-Depression Investigation of Cloud systems in the Tropics | 2010 | 10°N – 29°N, 87°W – 38°W | 92 |
| | START08 | Stratosphere-Troposphere Analyses of Regional Transport | 2008 | 26°N – 63°N, 117°W – 86°W | 55 |
| | TORERO | Tropical Ocean tRoposphere Exchange of Reactive halogen species and Oxygenated voc | 2012 | 42°S – 14°N, 105°W – 70°W | 54 |
| NASA | ATTREX | Airborne Tropical TRopopause EXperiment | 2014 | 12°S – 36°N, 134°E – 117°W | 128 |
| | MACPEX | Mid-latitude Airborne Cirrus Properties EXperiment | 2011 | 26°N – 41°N, 104°W – 84°W | 31 |
| | DC3 | Deep Convective Clouds and Chemistry Project | 2012 | 30°N – 42°N, 117°W – 106°W | 29 |
| | POSIDON | Pacific Oxidants, Sulfur, Ice, Dehydration, and cONvection | 2016 | 1°S – 15°N, 131°E – 161°E | 41 |
| | SEAC4RS | Studies of Emissions and Atmospheric Composition, Clouds and Climate Coupling by Regional Surveys | 2013 | 19°N – 50°N, 80°W – 120°W | 15 |

* Only used deployments #2 to #5.

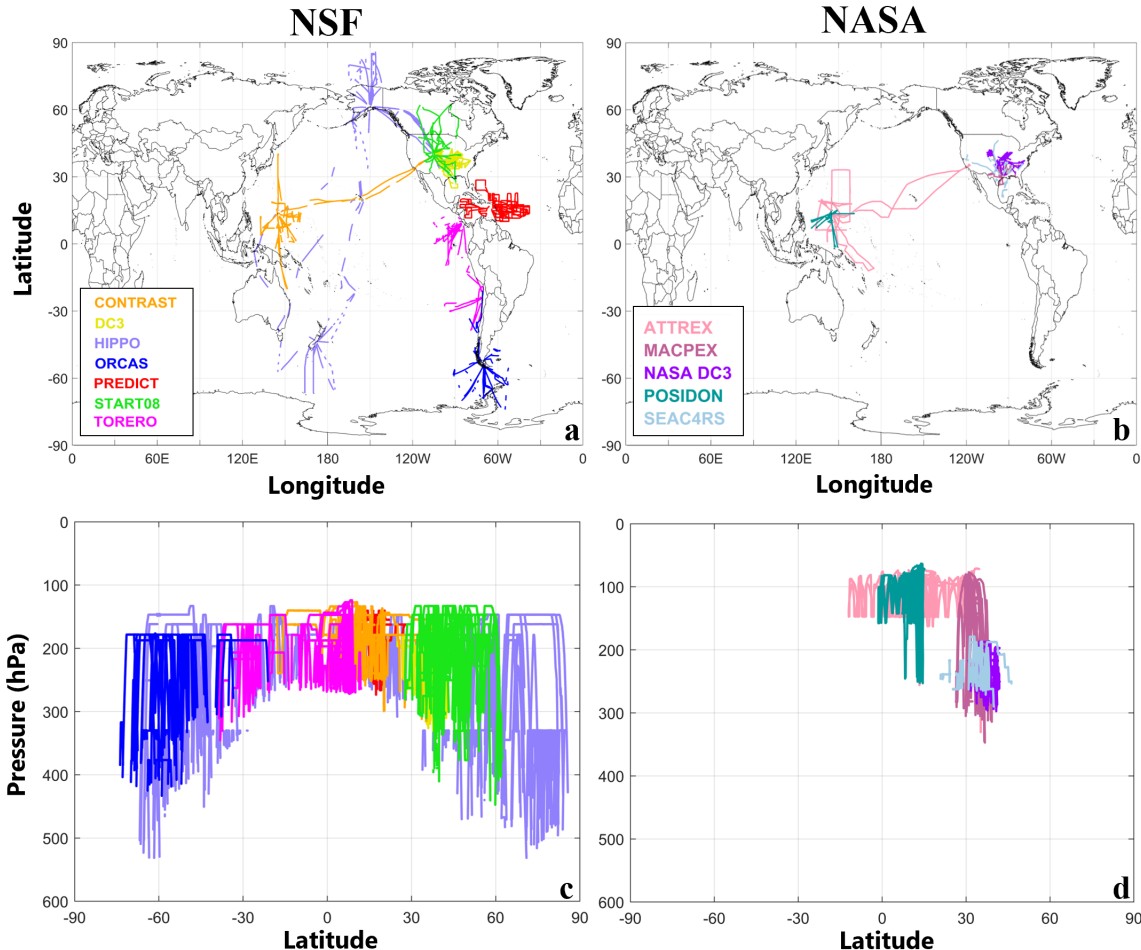

**Figure 1.** The flight tracks for (a) seven NSF flight campaigns and (b) five NASA campaigns. (c) and (d) show the latitudinal cross-section of the NSF and NASA datasets, respectively. All flight tracks are restricted to temperatures ≤ -40°C.

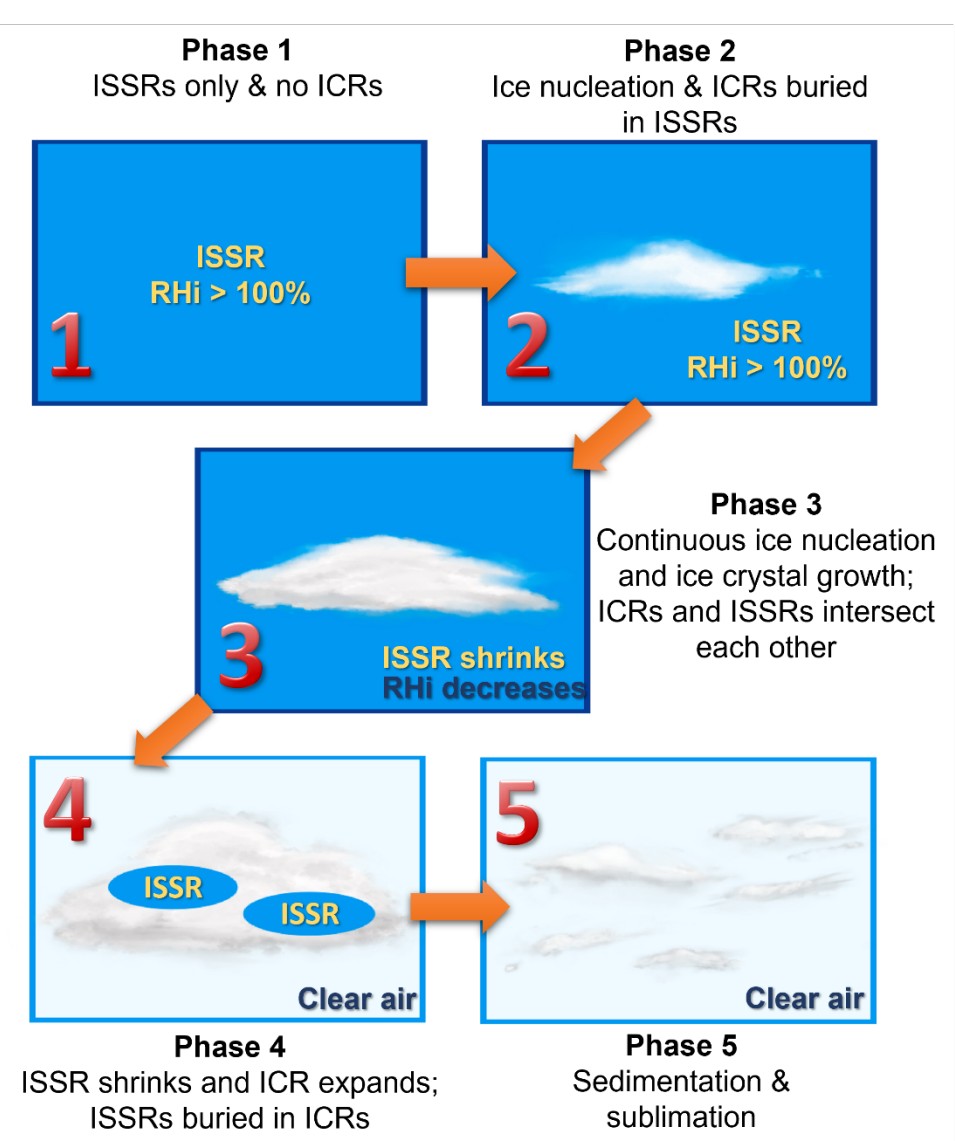

**Figure 2.** An idealized illustration of five cirrus cloud evolution phases as defined in Diao et al. (2013).

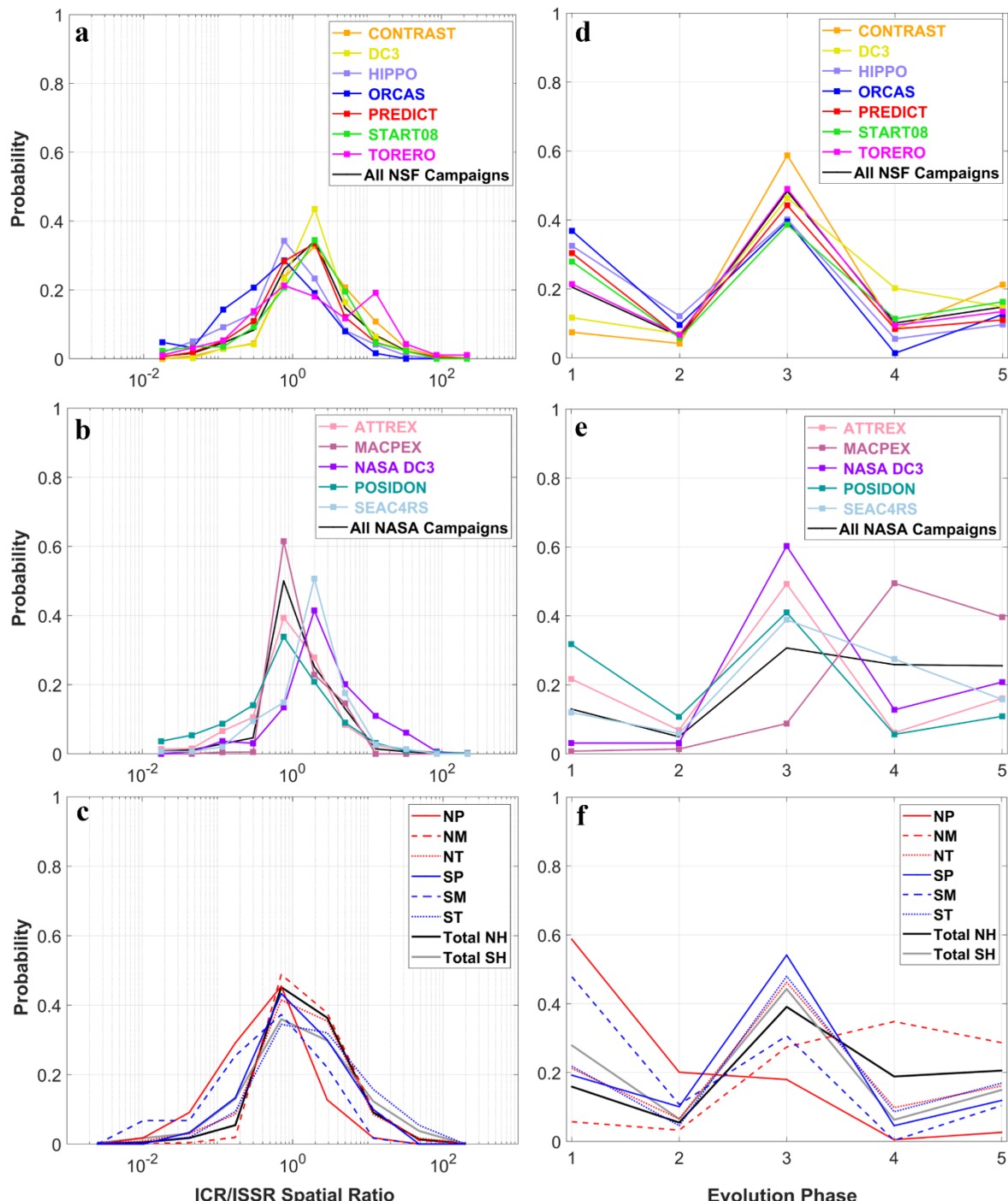

**Figure 3.** Probabilities of (a-c) the ICR/ISSR spatial ratio and (d-f) five evolution phases for NSF, NASA, and all campaigns in row 1, 2 and 3, respectively. The probabilities are shown for each campaign (in a, b, d and e) and for each latitudinal region (in c and f).

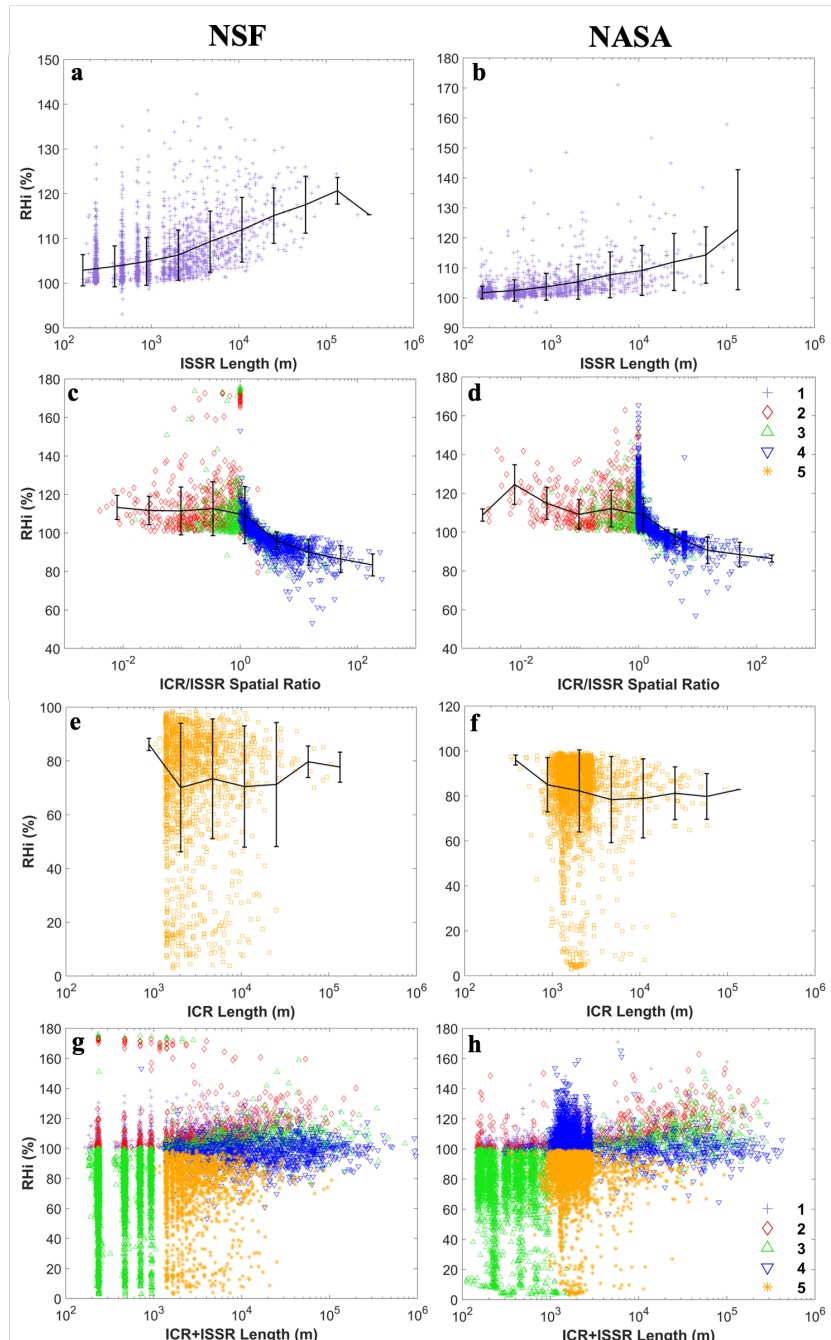

**Figure 4.** RHi distributions for (a, b) phase 1, (c, d) phases 2 to 4, (e, f) phase 5, and (g, h) all five evolution phases. Observations from NSF and NASA campaigns are shown in column 1 and 2, respectively. All error bars in this and the following figures represent ± one standard deviation.

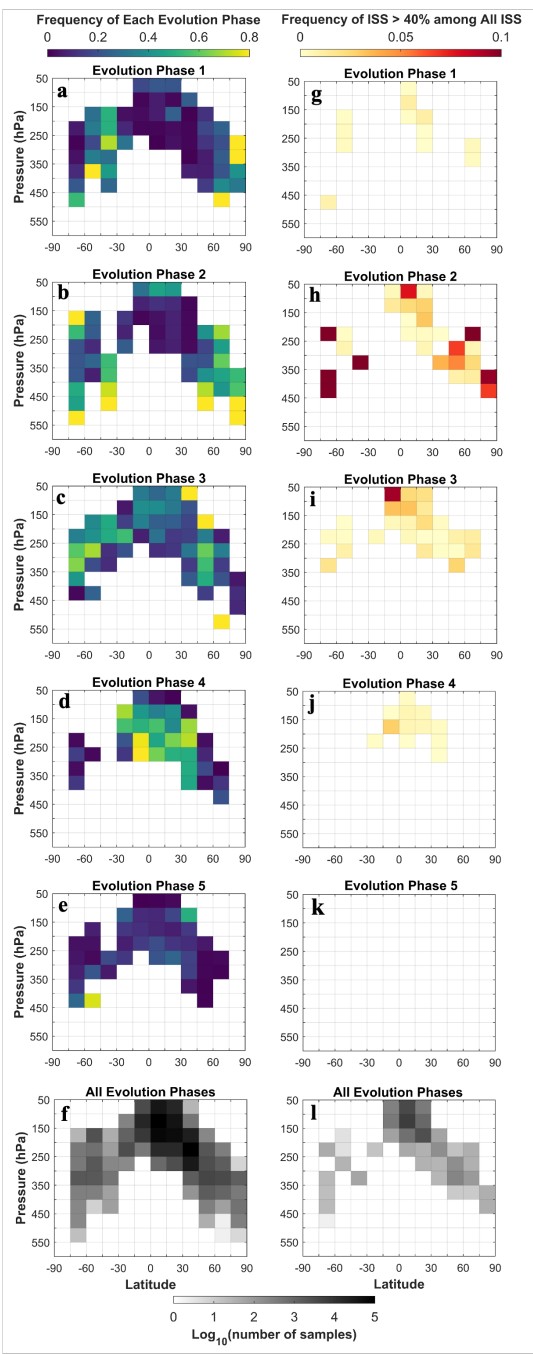

**Figure 5.** The hemispheric distribution of the occurrence frequencies of (a-e) each evolution phase and (g-k) RHi > 140% among all ice supersaturated conditions in each phase. (f) Number of samples for five evolution phases. (l) Number of samples for all ISS conditions in five phases. This analysis is based on combined NSF and NASA datasets.

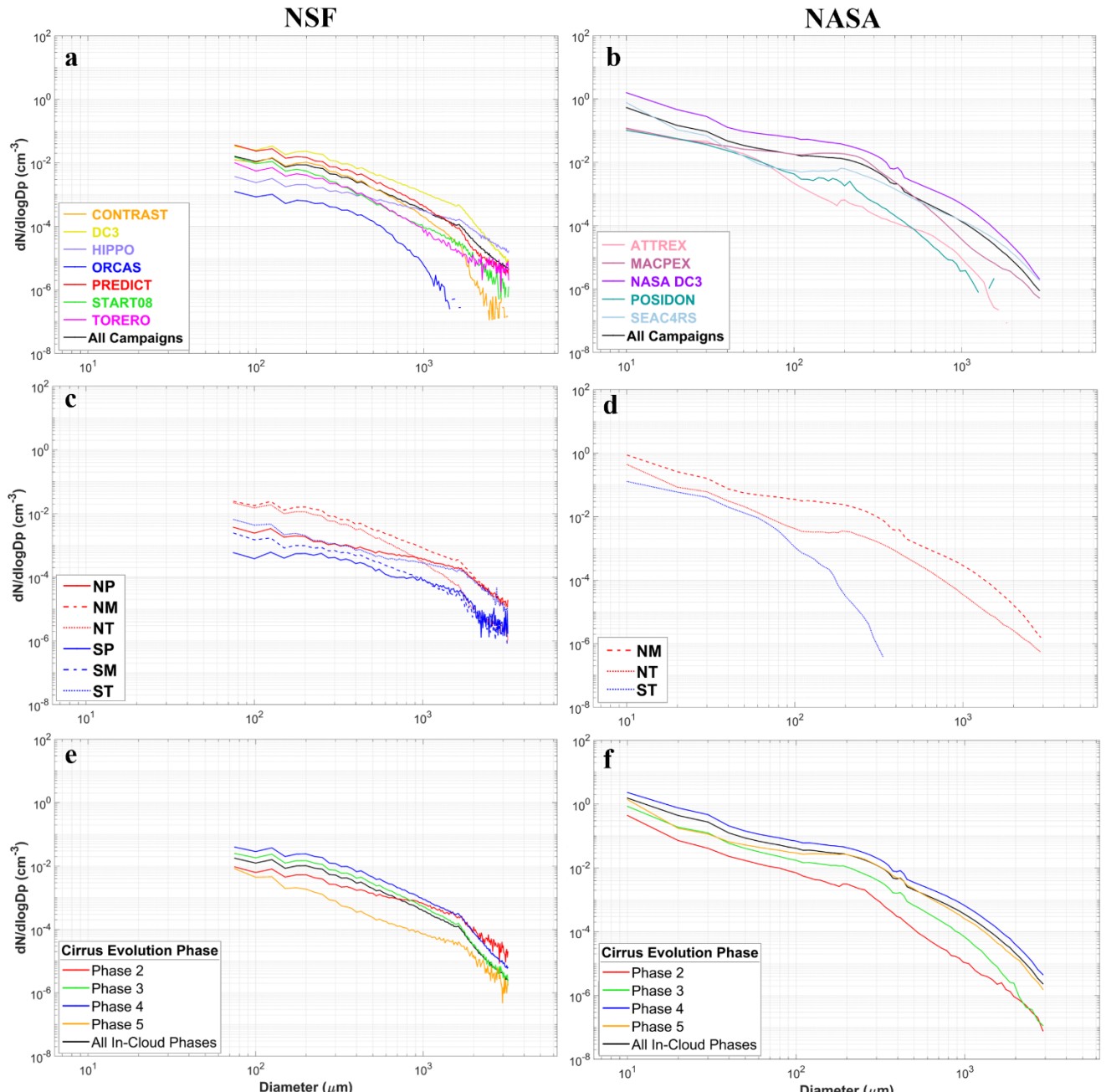

**Figure 6.** Particle size distribution plots using (a, c and e) the 2DC cloud probe for NSF campaigns and (b, d, and f) 2DS probe for NASA campaigns. The data are separated by (a, b) campaigns, (c, d) latitudinal regions, and (e, f) the cirrus cloud evolution phases. dN/dlogDp is the average particle number concentration in each bin normalized by the log10 scale bin width.

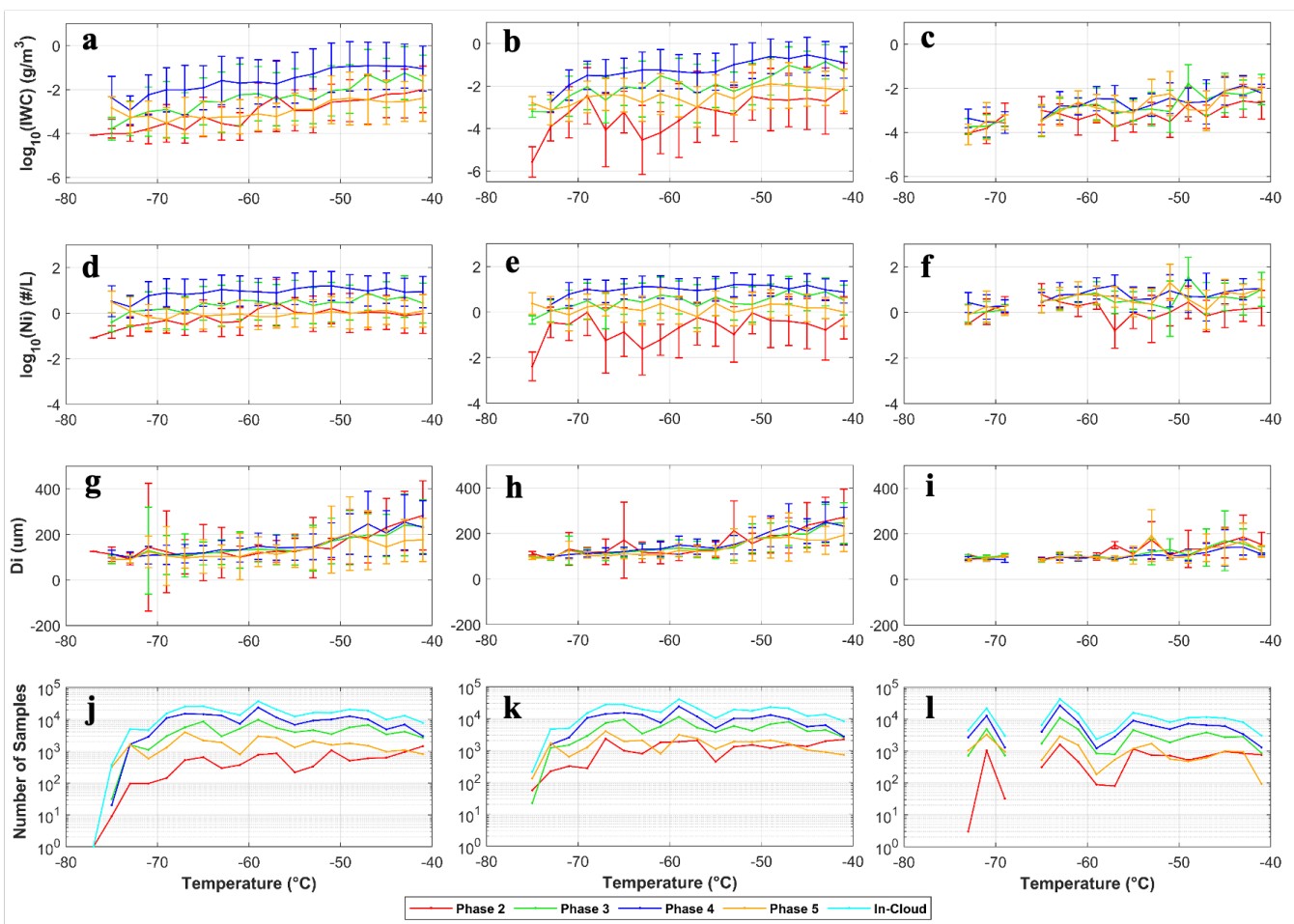

**Figure 7.** Averages of $\log_{10}$(IWC), $\log_{10}$(Ni), and Di in each 2-degree temperature bin for evolution phases 2 to 5 using 1-s observations, observations averaged to a 100-km horizontal scale, and the simulations in column 1, 2 and 3, respectively. The number of samples is shown in the last row. In Figures 7 – 11 and 14 – 17, only NSF campaigns are used for comparisons with model simulations since the NSF and NASA campaigns used cloud probes with different measurement ranges.

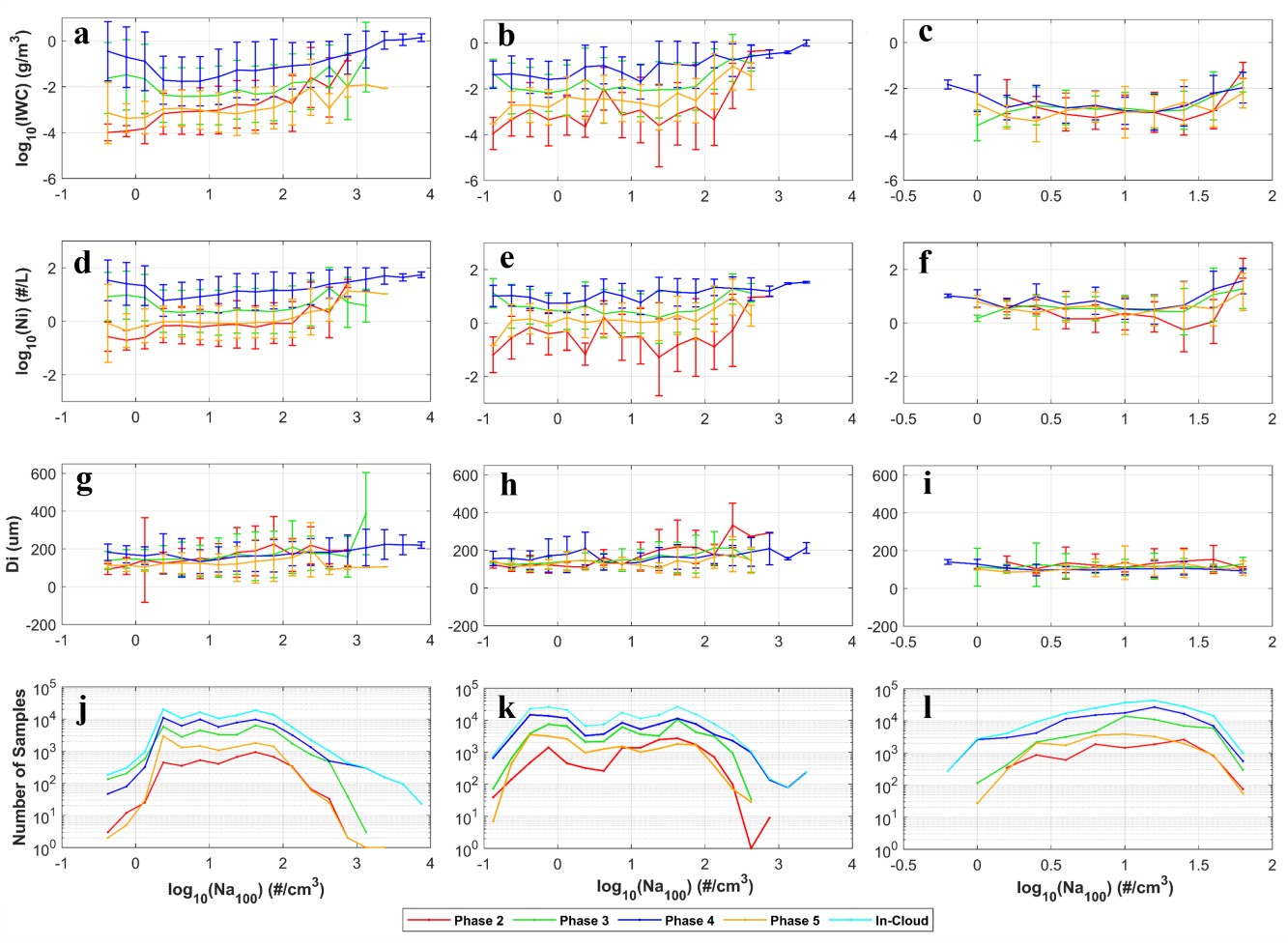

**Figure 8.** Similar to Figure 7, except for plotting cloud microphysical properties against $\log_{10}(Na_{100})$.

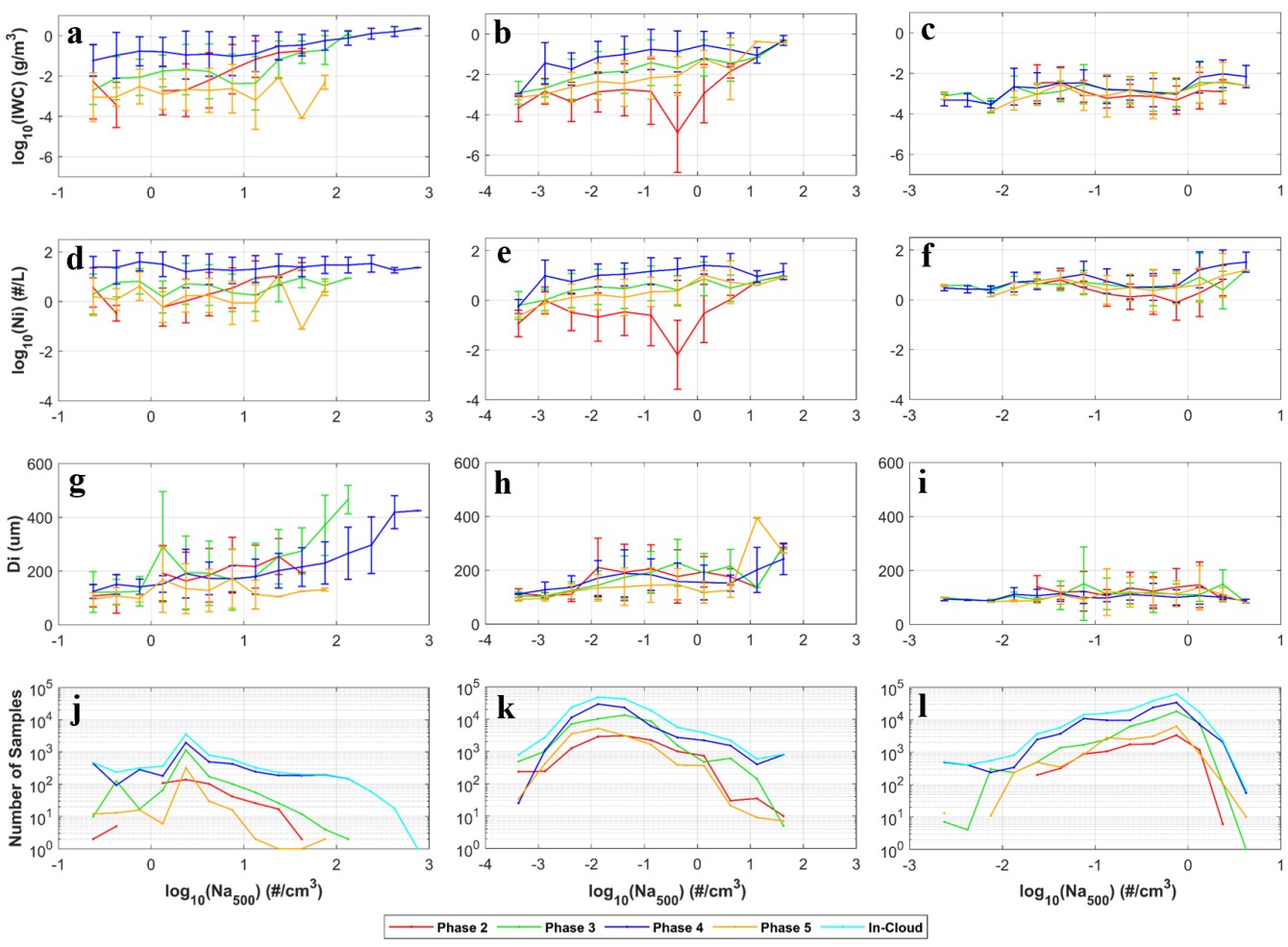


**Figure 9.** Similar to Figure 8, except for plotting cloud microphysical properties against $\log_{10}(Na_{500})$.

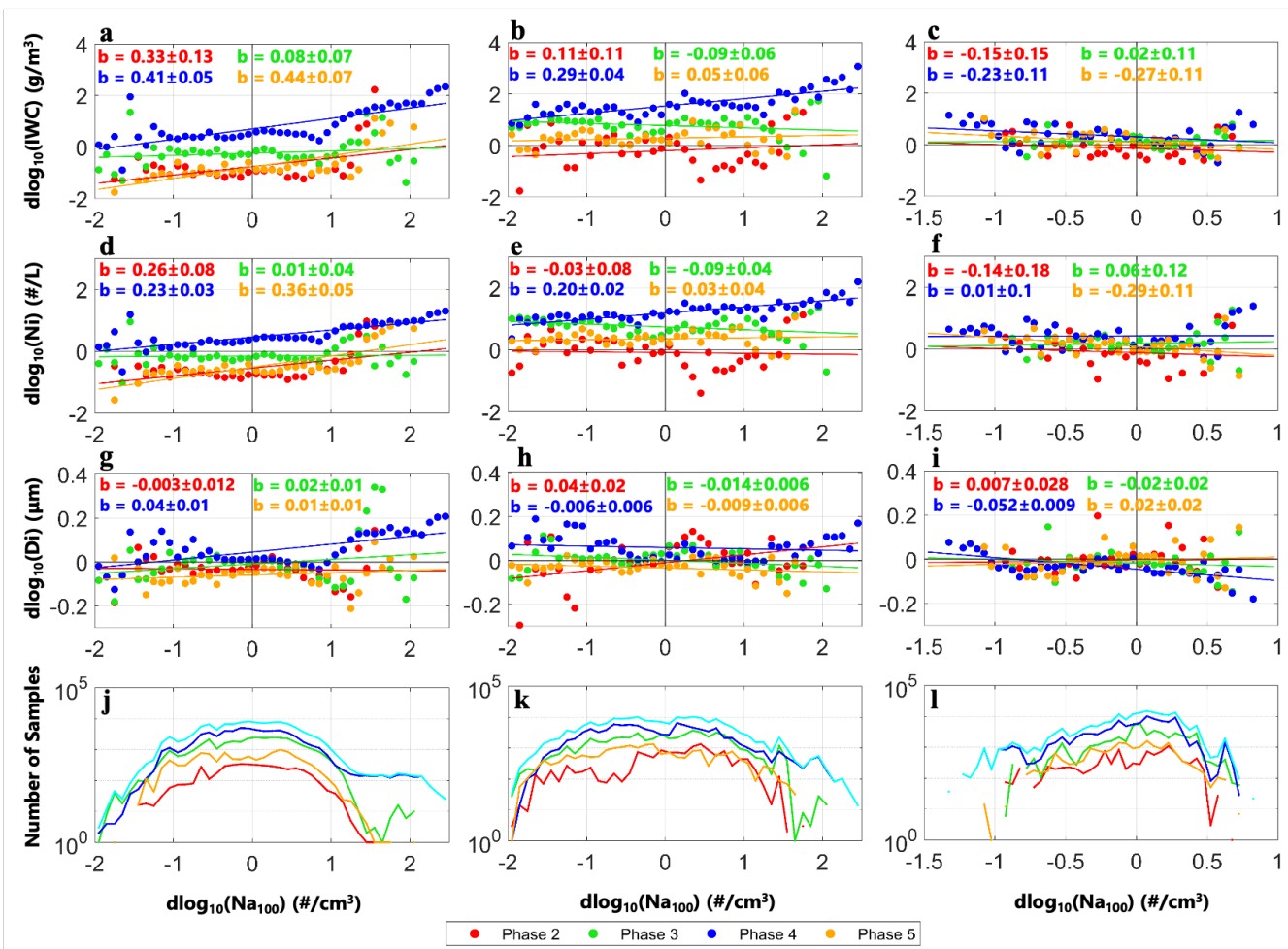

**Figure 10.** Linear regressions of the delta values of cirrus microphysical properties, i.e., $dlog_{10}(IWC)$, $dlog_{10}(Ni)$, and $dlog_{10}(Di)$ with respect to $dlog_{10}(Na_{100})$ for evolution phases 2 to 5 (various colored lines). Columns 1, 2 and 3 represent 1-s observations, 430-s observations, and simulations, respectively. The slope value and its standard deviation are shown for each linear regression. The number of samples is shown in the last row.

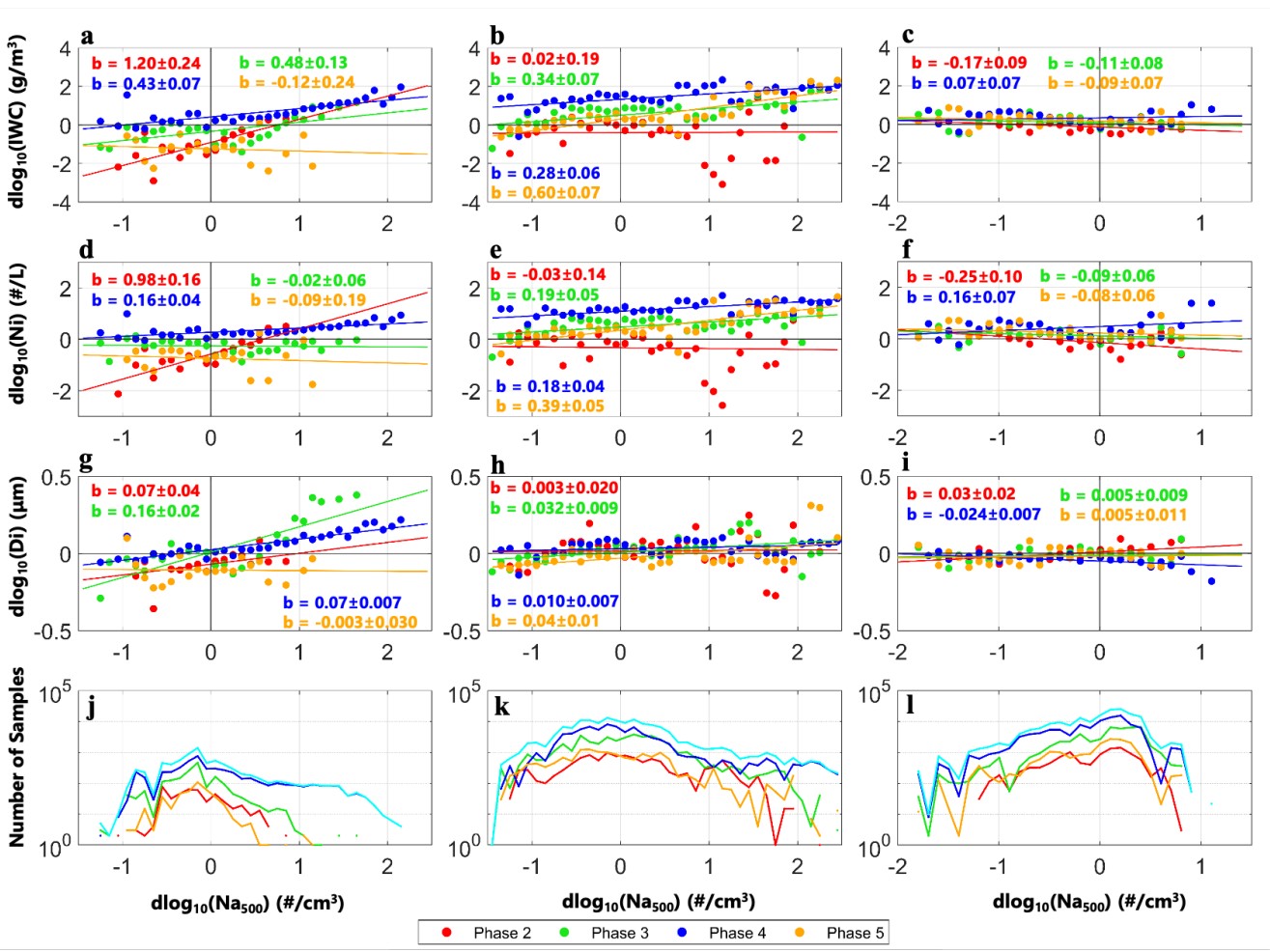

**Figure 11.** Similar to Figure 10, except for linear regressions with respect to $dlog_{10}(Na_{500})$.

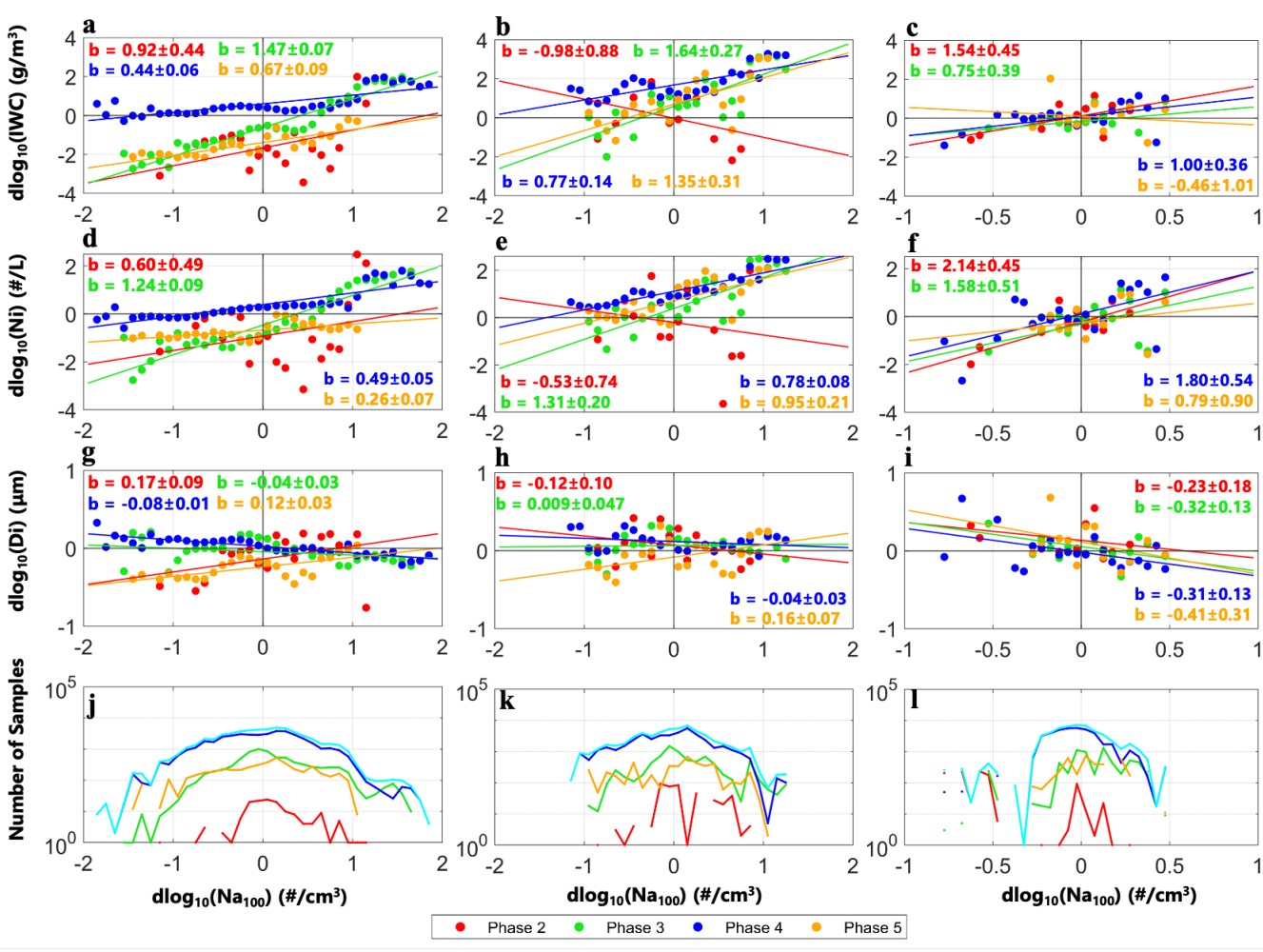

**Figure 12**. Similar to Figure 10, except for using two NASA flight campaigns to analyze linear regressions with respect to $dlog_{10}(Na_{100})$.


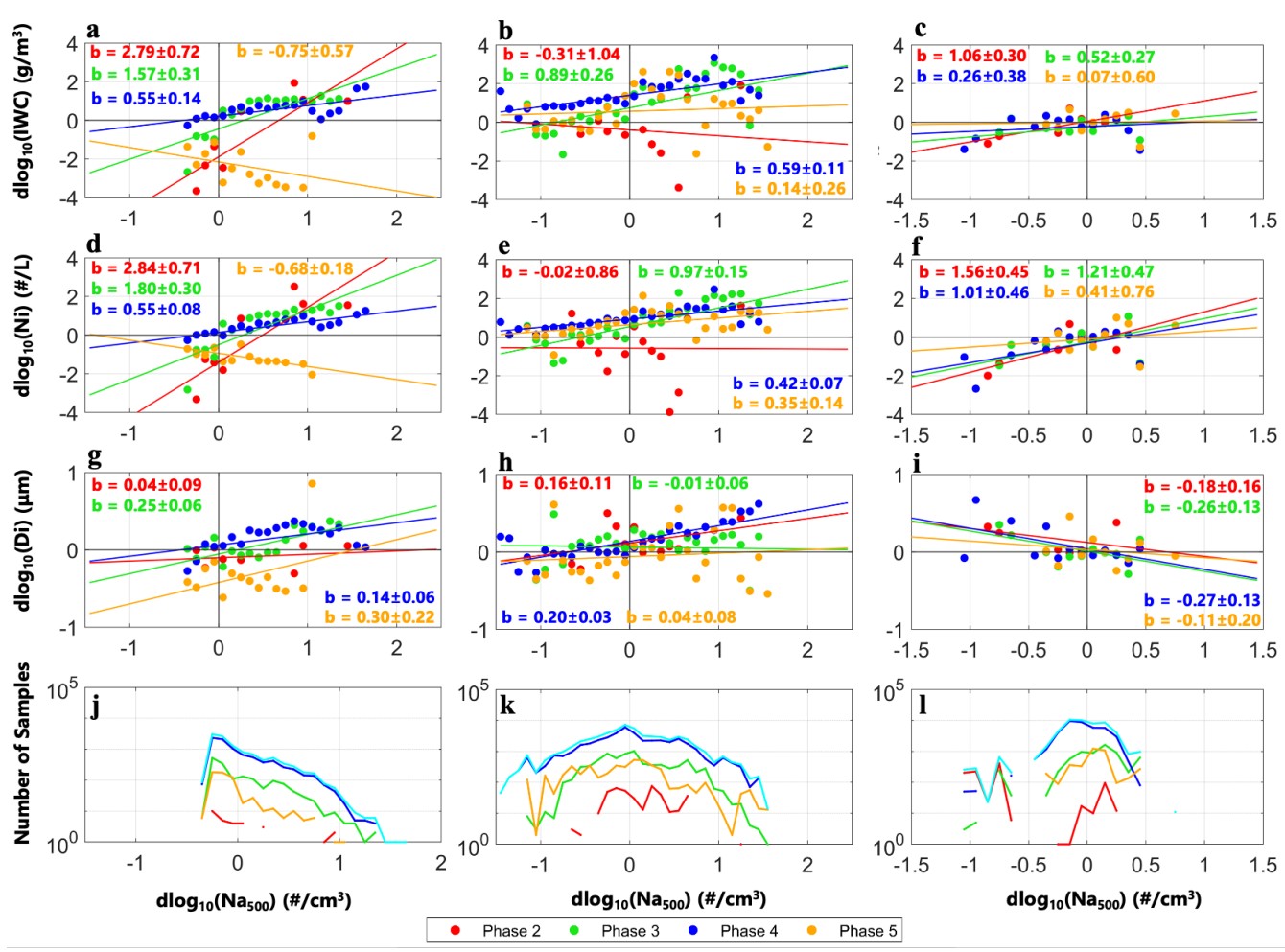

**Figure 13**. Similar to Figure 10, except for using two NASA flight campaigns to analyze linear regressions with respect to dlog$_{10}$(Na$_{500}$).

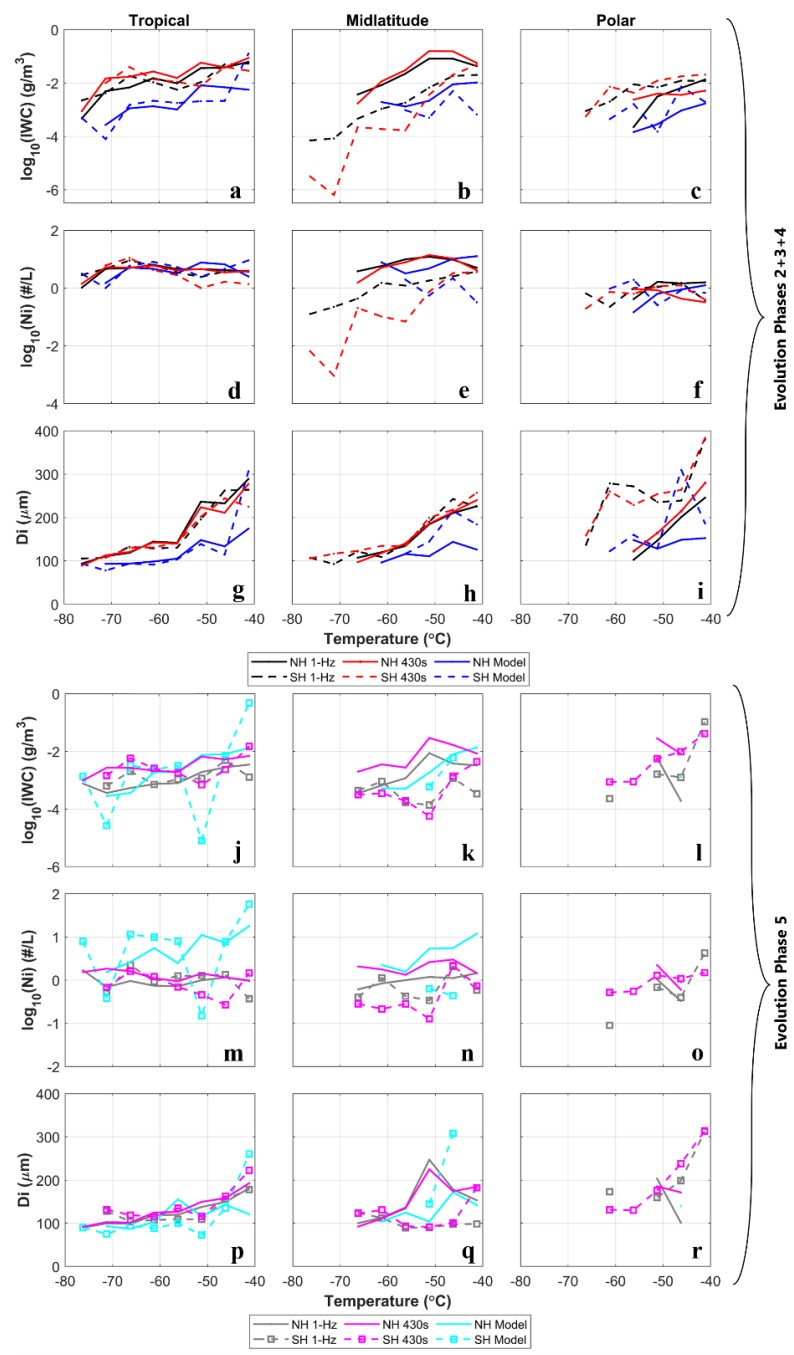

**Figure 14.** Averages of log$_{10}$(IWC), log$_{10}$(Ni), and Di for every 5-degree temperature bin for evolution phases 2 to 5, separated by NH and SH. The top three rows are for phases 2+3+4 while the bottom three rows are for phase 5 only. Columns 1, 2 and 3 represent tropical, midlatitudinal and polar regions, respectively. The number of samples is shown in the supplementary material (Figure S3).

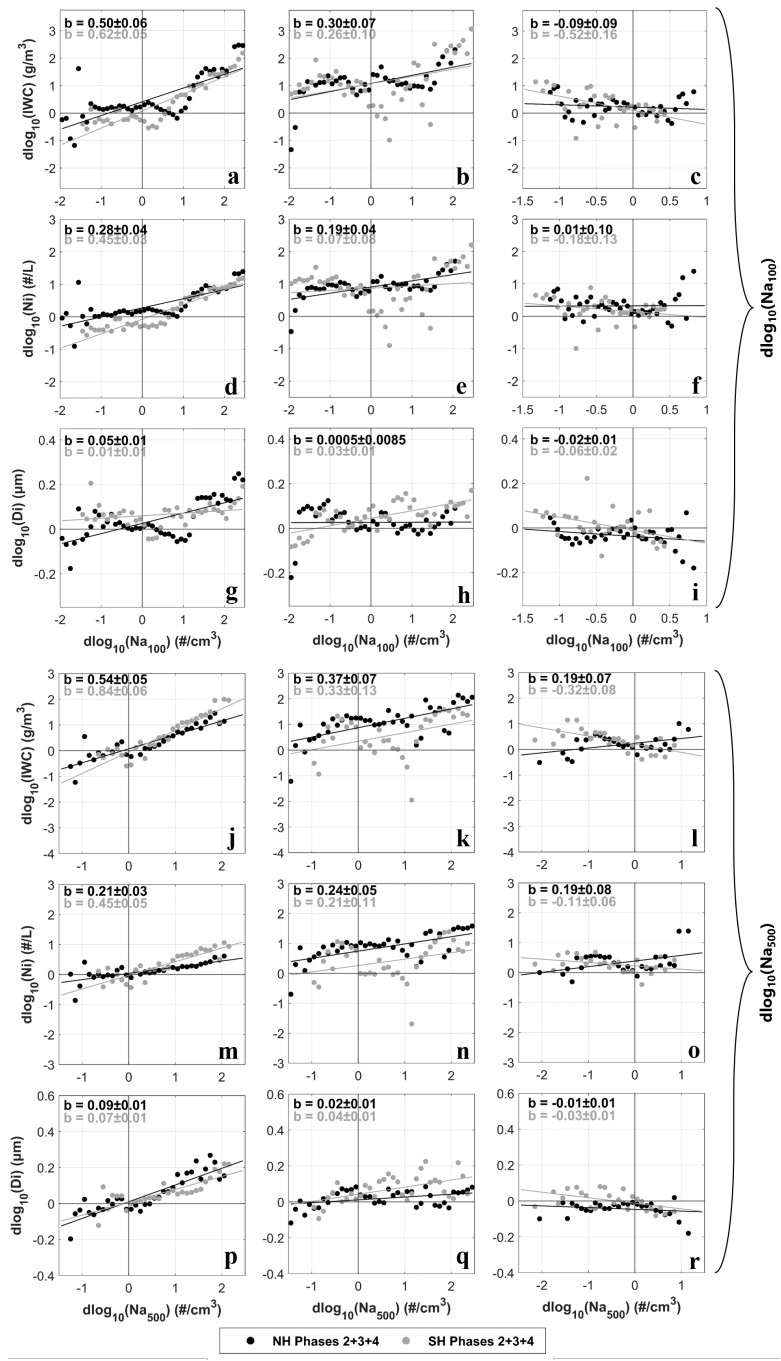

**Figure 15.** Linear regression of log₁₀(IWC), log₁₀(Ni), and Di with respect to the aerosol number concentrations, (top three rows) $Na_{100}$ and (bottom three rows) $Na_{500}$. This analysis shows the combined phases 2, 3 and 4, separated by NH and SH. Columns 1, 2 and 3 represent 1-Hz observations, 100-km scale observations and CAM6 simulations, respectively. The number of samples is shown in the supplementary material (Figure S4).

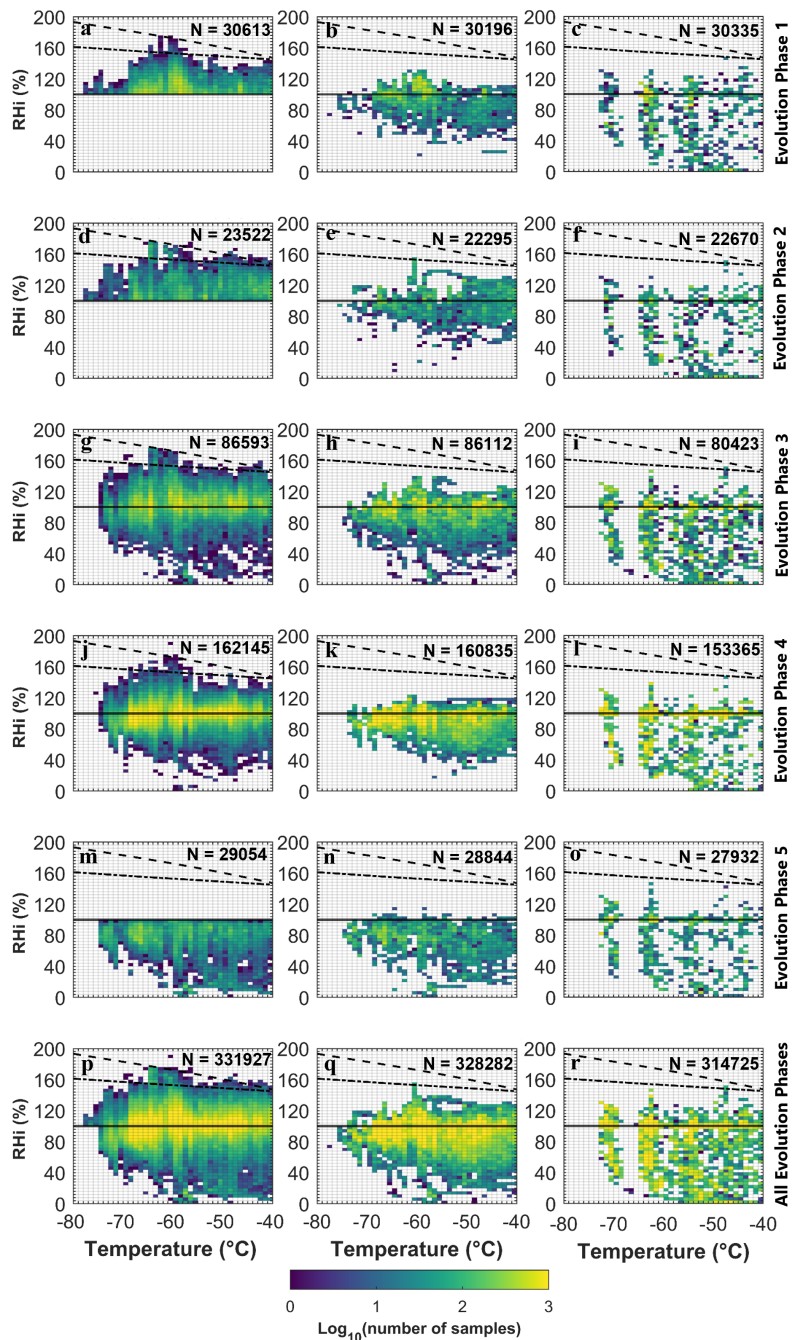


**Figure 16.** Distributions of RHi in relation to temperature for five evolution phases shown for 1-s observations, 430-s observations and simulations in column 1, 2 and 3, respectively. Number of samples (in seconds) is shown in logarithmic scale. The solid line represents ice saturation while the dashed line represents liquid saturation. The dot-dashed line shows the homogeneous nucleation threshold for aerosols with a size of 0.5 μm.

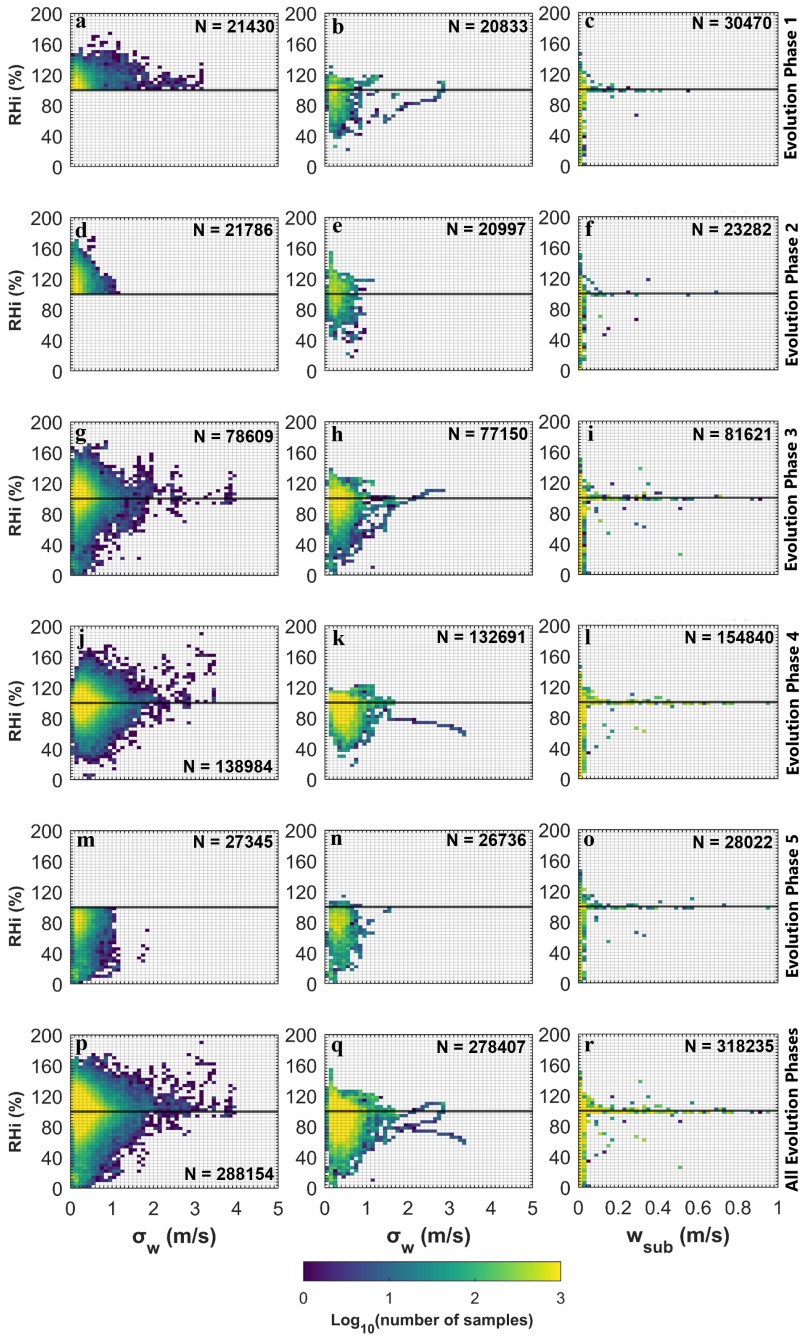

**Figure 17.** Similar to Figure 16, except for distributions of $\sigma_w$ in relation to temperature.