# Peer review of "Examination of Aerosol Indirect Effects during Cirrus Cloud Evolution"

_Atmospheric Chemistry and Physics, 2022_

## Referee Comment (RC1)

A review of 'Examination of Aerosol Indirect Effects during Cirrus Cloud evolution', by

Flor Vanessa Maciel,  Minghui Diao, Ryan Patnaude

submitted to Atmospheric Chemistry and Physics

This article reveals cirrus cloud microphysical properties and aerosol indirect effect in five cirrus evolution phases, i.e. clear-sky ice supersaturation region, nucleation, early growth of ice crystals, later growth of ice crystals and sedimentation/sublimation. The five phases were determined based on the method from Diao et al. 2013. Using 12 field campaign measurements, the authors investigated the five cirrus cloud evolution phases characteristics, and aerosol indirect effects in both northern and southern hemispheres. Some interesting results are found, including cirrus water content and ice crystal number concentration are positively correlated with aerosol number concentration – aerosol indirect effects are different for different size distributions of aerosol particles at different cirrus evolution phases, which is stronger for larger aerosol particles at nucleation phase, while it is stronger for later growth phase for smaller aerosols. The authors also indicate that aerosol indirect effect is stronger in southern hemisphere than in northern hemisphere. By comparing ice cloud microphysical properties and aerosol indirect effect with the simulations from NCAR CAM6, the authors found lower IWC and number concentrations compared with observations, and they hypothesized that lower frequencies of ice supersaturation and lower vertical velocity standard deviation in the early/later growth phases contribute to the underestimation of IWC and number concentration.

Overall, this paper is comprehensive. It provides some new thoughts of aerosol-cloud interaction from the Lagrangian view and possible solutions to improve model simulations. This paper deserves publication after some minor revisions.

My main comments are as follow:

1. In section 2, it is a little bit messy in describing the in-situ observations. I would suggest using a table to summarize the variables (e.g. IWC, Di, Ni, w, RHi, etc.), the instruments that the variables are from, references, and the variable uncertainties.
2. Also in section 2, the methods for data analysis are sometimes confusing. It is adopted from Diao et al. (2013), but it seems different in defining the 'consecutive sample'. Sometimes, the analysis is for pixel level. Making the definition clear is important to understand the results.
3. For aerosol indirect effect analysis, although the authors mentioned that the correlation or trend is significant, I do not see any significant tests for the regression. I would strongly suggest significant tests for the regression. Particularly, if the regression is insignificant, how to explain the results.
4. In Figures 8-11, Di varying with Na were shown, but there are not explanations for the results. The difference between Di vs Na100 and Di vs Na500 is obvious. Please describe these results in the text.

Specific comments:

Page 3, line 85-86: 'We applied the method of Diao et al. (2013) to derive various evolution phases of cirrus  clouds, which enables a detailed examination of aerosol indirect effect…'. I suggest describing

Diao et al. (2013) in more detail about their method of separating the five cirrus cloud evolution phases. Then in section 2, focused on your methods as a follow-up of Diao et al. (2013).

Page 3, Line 90: please specify 'Na'.

Page 4, Line 117 & 119: 'this merged observational dataset','For the merged datasets', what datasets?

Page 4, Line 119: 'For the merged datasets, ice supersaturated regions (ISSR) and ice crystal regions (ICR) were identified using values of RHi and Ni, respectively.' How the ISSR and ICR defined, in pixel-level, cloud-element level or in the same way as Diao et al. (2013)?

Page 6, Lines 170- 177: It is confusing about how to calculate the probability.  In line 171, it is mentioned that ' the result shows that a cloud segment has the highest probability ….', which indicates the results are for cloud segments. But in Line 173, it mentions that the method is counting points of observations, which is for pixel levels. Please clarify your methods in the method section.

Page 6, Line 186: 'NM and NT have lower frequencies of clear-sky ISSRs compared with their counterparts in the SH', NM and SM show large difference in the frequency, but NT and ST show very close frequency of phase 1. It is better to explain midlatitude and tropical regions separately.

Page 7, Line 189: frequency of Phase 1 is about 0.2 not 0.35.

Page 7, Line 196: 'reduces' → reduce

Page 7, Line 200: 'interestingly, the highest RHi values are mostly seen in phase2, …', this is a good point. The result here also agrees with Diao et al, 2013.

Figure 5: fig. k is confusing. Is it fig. l/ fig. f? Please describe it clearly.

Page 8, Line 227: 'compared with in-situ observations', in-situ formed cirrus clouds.

Page 8, Line 230-234: 'This is consistent with the fact that ATTREX mostly sampled the western Pacific Ocean region', This is not a good explanation for the particle number concentration between NASA DC3 and ATTREX.

Page 9, Lines 274-278: it is confusing to me how temperature affects the variation among the phases.

Page 9, Line 282: 'by subtracting the average values of the corresponding 1-degree temperature bin from the individual bin.', it is not clear how to deal with the temperature here.

Figures 14&15: please add colorbars.

Page 11, Line 343: please clarify what $w_{sub}$ represents.

Page 11, Line 348: how the $\sigma_w$ impact ice formation?

Line 355: 'RHi distributions with respect to temperature', do you show that in the paper?

---

## Referee Comment (RC2)

Review on "Examination of Aerosol Indirect Effects during Cirrus Cloud Evolution"
by Maciel et al.

Aerosol effects on clouds constitute of one of the largest uncertainties in climate change projection. Particularly, aerosol effects on cirrus clouds are rarely studies but can be potentially important. This study investigates the aerosol indirect effects on cirrus clouds by analyzing in situ aircraft observations of cloud microphysical properties and aerosol number concentrations from NSF and NASA campaigns. The uniqueness of this study is that the analysis separates out five evolution phase of cirrus clouds so that the aerosol indirect effects can be more robustly examined. The observational data analysis is also used to compare with and evaluate the CAM6 model simulations of these field campaigns. Overall, the analysis is solid and the results are convincing. The manuscript is well written. I recommend the publication of this manuscript in ACP after my comments are addressed.

My major comment is that an important objective of this study is to evaluate the model simulations with observation data and some outstanding biases of the model simulations are thus identified. Therefore, it would be helpful to add some descriptions of the CAM6 model parameterizations of cloud microphysics and aerosols. I have also some minor comments as outlined below.

1. Abstract. L19-21. "*Observations show stronger aerosol indirect effects (i.e., positive correlations between IWC, Ni and Na) in the Southern Hemisphere (SH) compared with the Northern Hemisphere (NH),..*" It is somehow count-intuitive because there are not as many INPs (e.g., dust) in SH as in NH. The authors are suggested to add some more elucidations for this statement.
2. L50. "*the number of ice crystals that are nucleated is affected by both strong diffusion and turbulence.*" The number of ice crystals that are nucleated is affected by cooling rate (or updrafts). Can the authors clarify what "diffusion" means here?
3. L132. What do you mean "center point phase identifications"?
4. L136. When you use NCAR CAM6 to conduct simulations it would be needed to add some descriptions of CAM6 parameterizations of cloud microphysics and aerosols.
5. L136. Here you mention the NSF campaigns. Why do not you conduct the similar simulations for the NASA campaigns?
6. L152. How do you calculate Na100 and Na500 from modeled aerosol modes?
7. L164. How do you calculate the ICRs and ISSRs spatial scales?
8. L171. What is the implication of this result? "*a cloud segment has the highest probability to be in the early growth phase (i.e., phase 3) for almost all the campaigns (except MACPEX).*"
9. L189. Shall "0.35" be "0.2"?
10. L266-270. "*This feature indicates that as Na500 exceeds a threshold, the available INPs in the air parcel may have been depleted, and therefore no new ice nucleation can be initiated with additional larger aerosols that are not INPs.*" Here, the explanation is vague and not very convincing. there is no data beyond logNa500> 1.3. Na500 cannot be depleted at those high concentrations (>10 cm$^{-3}$). What are the additional larger aerosols that are not INPs?

11. L326. "…*that increasing aerosol concentrations in the SH may be more effective in increasing ice nucleation compared with the NH*." This statement is interesting but may need more elucidation.
12. L331-332. Here the authors talk about the model bias related to cirrus microphysical properties. However, there is also large bias in aerosols (Na100 and Na500). Add model-observation comparisons of aerosols.
13. L342. Shall "40 seconds" be "1 second"?
14. L348. "Both the lower $\sigma_w$ values and lower ice supersaturation frequencies can contribute to the lower IWC in the model". How about modeled Ni?
15. L378. Here the authors talk about the model sub-grid scale supersaturation and w and their impact on IWC. Why is the modeled Ni relatively better simulated than IWC?

---

## Author Response (AR1)

**Overarching Responses to the Reviewers**

Format: The reviewer's comments are quoted in italic

Line number in the response refers to the revised manuscript with tracked changes

Quotation in red color stands for revised/added text in the revised manuscript

We thank the reviewers for the helpful comments. To sum up the main changes to the figures and tables, we added new Figures 12 and 13 (figure number refers to the revised manuscript) and Figure S2 – S4, S6, S9, S10 to compare the NASA campaign observations with CAM model simulations. In addition, we added statistical significance information to supplemental Table S3 – S5, added new Table S1 to summarize instrumentation and variable uncertainty, and added new Figure S1, which is a schematic diagram to clarify identifications of cirrus evolution phases. For minor changes, we updated several figures by adding the missing color bars and revising the color scheme to be reader friendly.

Below is our response to each of the comments, and the revisions to the main text accordingly.

**Response to comments from Reviewer 1**

*A review of 'Examination of Aerosol Indirect Effects during Cirrus Cloud evolution', by*

*Flor Vanessa Maciel, Minghui Diao, Ryan Patnaude*

*submitted to Atmospheric Chemistry and Physics*

*This article reveals cirrus cloud microphysical properties and aerosol indirect effect in five cirrus evolution phases, i.e. clear-sky ice supersaturation region, nucleation, early growth of ice crystals, later growth of ice crystals and sedimentation/sublimation. The five phases were determined based on the method from Diao et al. 2013. Using 12 field campaign measurements, the authors investigated the five cirrus cloud evolution phases characteristics, and aerosol indirect effects in both northern and southern hemispheres. Some interesting results are found, including cirrus water content and ice crystal number concentration are positively correlated with aerosol number concentration – aerosol indirect effects are different for different size distributions of aerosol particles at different cirrus evolution phases, which is stronger for larger aerosol particles at nucleation phase, while it is stronger for later growth phase for smaller aerosols. The authors also indicate that aerosol indirect effect is stronger in southern hemisphere than in northern hemisphere. By comparing ice cloud microphysical properties and aerosol indirect effect with the simulations from NCAR CAM6, the authors found lower IWC and number concentrations compared with observations, and they hypothesized that lower frequencies of ice supersaturation and lower vertical velocity standard deviation in the early/later growth phases contribute to the underestimation of IWC and number concentration.*

*Overall, this paper is comprehensive. It provides some new thoughts of aerosol-cloud interaction from the Lagrangian view and possible solutions to improve model simulations. This paper deserves publication after some minor revisions.*

*My main comments are as follow:*

*1. In section 2, it is a little bit messy in describing the in-situ observations. I would suggest using a table to summarize the variables (e.g. IWC, Di, Ni, w, RHi, etc.), the instruments that the variables are from, references, and the variable uncertainties.*

We thank the reviewer for the suggestions. A new **Table S1** (below) has been added to the Supplemental material to summarize the instruments, acronym, accuracy and precision, and measurement range.

**Table S1**. A description of instruments, their accuracy, precision, measurement range and related variables.

| Agency | Instrument | Accuracy and Precision | Measurement Range | Related Variables |
|---|---|---|---|---|
| NSF | Rosemount Temperature Probe [a,b] | ±0.3 K and 0.01 K | −80°C to +40°C | Temperature, RHi |
| | Fast 2-Dimensional Cloud Probe (Fast-2DC) [c] | 25 μm (pixel size) | 62.5 – 3200 μm | IWC, Ni, Di |
| | Vertical Cavity Surface-Emitting Laser (VCSEL) Hygrometer [d] | ~6% and ≤1% | −85°C to +32°C frost/dewpoint temperature | RHi |
| | Ultra-High Sensitivity Aerosol Spectrometer (UHSAS) [e] | 5% and 2.5% | 0.060 – 1.0 μm | $Na_{100}$, $Na_{500}$ |
| NASA | Meteorological Measurement System (MMS) [f,g] | ±0.3 K and ± 0.05 K | −90°C to +40°C | Temperature, RHi |
| | 2D-Stereo (2-DS) Probe [h,i] | 10 μm (pixel size) | 5 – 3005 μm | IWC, Ni, Di |
| | Diode Laser Hygrometer (DLH) [j,k] | 5% (or 0.5 ppmv) and 0.5% (or 0.05 ppmv) | 1 – 50000 ppmv | RHi |
| | Harvard Lyman-α Photofragment Fluorescence Water Vapor (HWV) Hygrometer* [l,m] | 5% and 1% | 1 – 1000 ppmv | RHi |
| | Ultra-High Sensitivity Aerosol Spectrometer (UHSAS) [n,o] | 5% and 2.5% | 0.060 – 1.0 μm | $Na_{100}$, $Na_{500}$ |

\* HWV was used for NASA MACPEX campaign only.

Sources for instrument accuracy, precision and measurement range in Table S1 are listed below.

a: Temperature sensor, UCAR, https://www.eol.ucar.edu/instruments/high-rate-ambient-temperature-sensor

b: Temperature sensor, UCAR, https://www.eol.ucar.edu/instruments/heated-ambient-temperature-sensor
c: Fast 2DC probe, UCAR, https://www.eol.ucar.edu/instruments/two-dimensional-optical-array-cloud-probe
d: Zondlo et al. Vertical cavity laser hygrometer for the National Science Foundation Gulfstream-V aircraft, JGR Atmosphere, 115, D20309, doi:10.1029/2010JD014445, 2010.
e: UHSAS instrument, NCAR, https://www.eol.ucar.edu/instruments/ultra-high-sensitivity-aerosol-spectrometer
f: MMS system, NASA,
https://airbornescience.nasa.gov/mms/content/METEOROLOGICAL_MEASUREMENT_SYSTEM
g: Scott et al. The Meteorological Measurement System on the NASA ER-2 Aircraft, Journal of Atmospheric and Oceanic Technology, 525-540, https://doi.org/10.1175/1520-0426(1990)007<0525:TMMSOT>2.0.CO;2, 1990.
h: 2DS probe, NASA, https://airbornescience.nasa.gov/instrument/2DS
i: Lawson et al. The 2DS (Stereo) Probe: Design and preliminary tests of a new airborne, high speed, high-resolution particle imaging probe, J. Atmos. Oceanic Technol., 23, 1462-1477, 2006.
j: DLH hygrometer, NASA, https://airbornescience.nasa.gov/instrument/DLH
k: Podolske et al. Calibration and data retrieval algorithms for the NASA Langley/Ames Diode Laser Hygrometer for the NASA Transport and Chemical Evolution Over the Pacific (TRACE-P) mission, J. Geophys. Res., 108, 8792, doi:10.1029/2002JD003156, D20, 2003.
l: Harvard Water Vapor Hygrometer, NASA, https://airbornescience.nasa.gov/instrument/HWV-LYA
m: Harvard Water Vapor Hygrometer, NASA,
https://airbornescience.nasa.gov/sites/default/files/documents/H2Ov_SEAC4RS.pdf
n: UHSAS instrument, NASA, https://airbornescience.nasa.gov/instrument/UHSAS
o: Cai et al. Performance characteristics of the ultra high sensitivity aerosol spectrometer for particles between 55 and 800 nm: Laboratory and field studies, J. Aerosol Sci., 39, 759-769, 2008.

*2. Also in section 2, the methods for data analysis are sometimes confusing. It is adopted from Diao et al. (2013), but it seems different in defining the 'consecutive sample'. Sometimes, the analysis is for pixel level. Making the definition clear is important to understand the results.*

This is a very good point. We added more clarification on the similarities and differences with respect to the original method of Diao et al. (2013).

Line 143: "For the 1-Hz observation dataset, ice supersaturated regions (ISSR) and ice crystal regions (ICR) were identified using values of RHi and Ni, respectively. ISSRs are regions where 1-Hz RHi is consecutively above 100%. ICRs are regions with consecutive in-cloud conditions at 1 Hz."

Line 151: "The definitions of ISSR, ICR, and five evolution phases follow the same criteria as those described in Diao et al. (2013). The difference between this study and Diao et al. (2013) is that the previous study only analyzed the averaged conditions (such as average RHi, IWC, Ni, and Di) for the entire evolution phase segment, while this study not only analyzes the average conditions (i.e., Figure 4) but also analyzes every second of measurements inside a phase segment by labelling each second with the phase number where that second belongs to (i.e., Figures 5 – 17). This means that in Diao et al. (2013), only one data point was used to represent one phase segment even if that segment contains many seconds of data, while this study analyzes every 1-Hz datum within each phase, which significantly increases the sample size. An example diagram illustrating the differences between this study and Diao et al. (2013) is shown in supplemental Figure S1."

Below is the new supplemental Figure S1.

[Figure]

**Figure S1**. An example illustrating the similarities and differences between this study and Diao et al. (2013). A segment of evolution phase 3 is defined for the intersection of ISSR and ICR. This phase definition is the same between the two studies. The only difference is that Diao et al. (2013) analyzed the average condition of this segment, while this study analyzes each second within this segment, all labelled as phase 3, except for Figure 4 which uses the segment-average RHi.

*3. For aerosol indirect effect analysis, although the authors mentioned that the correlation or trend is significant, I do not see any significant tests for the regression. I would strongly suggest significant tests for the regression. Particularly, if the regression is insignificant, how to explain the results.*

This is a good suggestion. We added two types of significant tests to the analysis, including the $R^2$ value and p-value. The new information is shown in supplemental Tables S3 – S5.

For example, we refer to Table S3 for statistical significance test for Figures 10 and 11: "Table S3 in the Supplement documents the full linear regression equations, including slopes, intercepts, and their standard deviations. Indicators of statistical significance such as $R^2$ values and p-values are also shown in that table."

We also added comments on the $R^2$ values in Line 414: "The dominance of heterogeneous freezing on the nucleation phase can be seen from the slope values of IWC and Ni, i.e., 1.2 and 0.98 ($R^2 = 0.61$ and 0.70) with respect to $Na_{500}$, which are about 3.6 – 3.8 times of the slope values of 0.33 and 0.26 ($R^2 = 0.19$ and 0.27) with respect to $Na_{100}$, respectively. The higher $R^2$ values for $Na_{500}$ also indicate higher statistical significance for the correlations with $Na_{500}$."

*4. In Figures 8-11, Di varying with Na were shown, but there are not explanations for the results. The difference between Di vs Na100 and Di vs Na500 is obvious. Please describe these results in the text.*

We added discussions on Di for Figures 8 – 11 in the following text.

Discussion on Figures 8 and 9 in Line 361: "For the mean diameter Di, a slight increase of Di is seen with increasing $Na_{100}$ and $Na_{500}$. The increase of Di is more significant with increasing $Na_{500}$ than $Na_{100}$, especially for phases 2 and 3. The average Di for five phases is mostly at or below 200 µm when examined against various $Na_{100}$ values. In contrast to that, the average Di exceeds 200 µm when $Na_{500}$ exceeds 10 cm$^{-3}$ for phases 2–4. The phases 3 and 4 even reach Di around 450 µm at higher $Na_{500}$. This further indicates that $Na_{100}$ are more likely correlated with homogeneous freezing, which produces

smaller ice crystals, while $Na_{500}$ are more correlated with heterogeneous nucleation, which produces larger ice crystals."

Discussion on Figures 10 and 11 in Line 411: "Contrasting the role of smaller and larger aerosols based on high-resolution observations, the smaller aerosols show the strongest positive correlations with IWC, $Ni$ and $Di$ in phase 4. … In addition, the nucleation phase shows higher slope value of $Di$ with respect to $Na_{500}$ (i.e., 0.07), while the slope is negative (-0.003) with respect to $Na_{100}$."

*Specific comments:*

*Page 3, line 85-86: 'We applied the method of Diao et al. (2013) to derive various evolution phases of cirrus clouds, which enables a detailed examination of aerosol indirect effect…'. I suggest describing Diao et al. (2013) in more detail about their method of separating the five cirrus cloud evolution phases. Then in section 2, focused on your methods as a follow-up of Diao et al. (2013).*

We added more clarification in Line 99: "The definition of the five evolution phases is identical to the method of Diao et al. (2013). While that former study only analyzed the averaged values for each phase segment, this study mostly focuses on 1-Hz samples within each phase segment." Because the definition of five evolution phases in this study is the same as Diao et al. (2013), we explain the details of their definition as we describe Figure 2 (the diagram of five phases) in Section 2.1 instead of addressing them inside the introduction.

We also addressed this question in our responses above, added descriptions to Section 2.1 and added supplementary Figure S1.

*Page 3, Line 90: please specify 'Na'.*

We added the definition of Na in Line 104: "…aerosols number concentrations (Na)…".

*Page 4, Line 117 & 119: 'this merged observational dataset','For the merged datasets', what datasets?*

We revised this sentence to: "In addition, we applied extensive quality control to this observational  dataset of multiple NSF and NASA campaigns." We deleted the word "merged".

*Page 4, Line 119: 'For the merged datasets, ice supersaturated regions (ISSR) and ice crystal regions (ICR) were identified using values of RHi and Ni, respectively.' How the ISSR and ICR defined, in pixel-level, cloud-element level or in the same way as Diao et al. (2013)?*

We clarified this question in our responses above. In short, ISSRs and ICRs are defined in the same way as Diao et al. (2013) as described in Line 143: "For the 1-Hz observation dataset, ice supersaturated regions (ISSR) and ice crystal regions (ICR) were identified using values of RHi and Ni, respectively. ISSRs are regions where 1-Hz RHi is consecutively above 100%. ICRs are regions with consecutive in-cloud conditions at 1 Hz."

*Page 6, Lines 170- 177: It is confusing about how to calculate the probability. In line 171, it is mentioned that ' the result shows that a cloud segment has the highest probability ....', which indicates the results are for cloud segments. But in Line 173, it mentions that the method is counting points of observations, which is for pixel levels. Please clarify your methods in the method section.*

Thank you for pointing this out. The original description was not accurate since the probability of panel a-c is calculated differently from the probability in panel d-f in Figure 3.

We added these clarifications:

Line 218: "The probabilities of ICR/ISSR spatial ratios for each campaign and various latitudinal regions are shown in Figure 3 a – c. This probability is based on cloud segment number, which is calculated as the number of cloud segments in a bin of $\log_{10}$(ICR/ISSR) spatial ratio divided by the total number of cloud segments with real values of $\log_{10}$(ICR/ISSR) in all bins, similar to the calculation in Diao et al. (2013) and their Figure 4 b."

Line 228: "Five evolution phases are identified for all flight campaigns, and their probabilities are shown in Figure 3 d – f. This probability of each phase is based on cloud segment length, which is calculated as the lengths of a certain phase divided by the total lengths of all five phases. This is different from Diao et al. (2013) and their Figure 4 a, which calculated the phase probability as the number of segments in a phase divided by the total number of all segments in five phases (i.e., number-based instead of length-based). The result shows that the early growth phase (i.e., when ISSR and ICR intersect each other) has the most dominant spatial coverage for almost all the campaigns (except MACPEX). Since the ICR/ISSR ratio is also around unity, these two results indicate that the coexistence of ISSRs and ICRs at similar spatial scales provides a semi-steady state for cirrus evolution, possibly because new ice crystal formation in ISSRs can balance out the sedimentation of aged ice crystals and therefore cirrus clouds can persist in this condition."

*Page 6, Line 186: 'NM and NT have lower frequencies of clear-sky ISSRs compared with their counterparts in the SH', NM and SM show large difference in the frequency, but NT and ST show very close frequency of phase 1. It is better to explain midlatitude and tropical regions separately.*

We agree with the suggestion and revise this sentence in Line 252 to: "Examining each latitudinal region between the two hemispheres, tropical regions show similar frequencies of each evolution phase between NT and ST. For the midlatitudes, NM shows lower frequencies of clear-sky ISSR and nucleation phases, and higher frequencies of later growth and sedimentation phase compared with SM. These results indicate that higher aerosol loading in the NH midlatitude may facilitate ice crystal formation and provide a faster transition from clear-sky ISSR and nucleation phases to early/later growth and sedimentation."

*Page 7, Line 189: frequency of Phase 1 is about 0.2 not 0.35.*

We thank the reviewer for catching this typo. We revised it to 0.2.

*Page 7, Line 196: 'reduces' → reduce*

Revised.

*Page 7, Line 200: 'interestingly, the highest RHi values are mostly seen in phase2, ...', this is a good point. The result here also agrees with Diao et al, 2013.*

We thank the reviewer for pointing out this similarity in results. We added a sentence to describe this: "This feature is also seen in the study of Diao et al. (2013)."

*Figure 5: fig. k is confusing. Is it fig. l/ fig. f? Please describe it clearly.*

We thank the reviewer for catching this typo and revised the legend in Figure 5.

*Page 8, Line 227: 'compared with in-situ observations', in-situ formed cirrus clouds.*

Thanks for the suggestion. We revised this sentence as suggested by the reviewer.

*Page 8, Line 230-234: 'This is consistent with the fact that ATTREX mostly sampled the western Pacific Ocean region', This is not a good explanation for the particle number concentration between NASA DC3 and ATTREX.*

We revise this part in Line 304: "Among all NASA campaigns, NASA DC3 has the highest particle number concentration per bin while ATTREX and POSIDON have the lowest. This may be caused by several reasons, e.g., different geographical locations (tropics versus midlatitude, land versus ocean) and different types of cirrus (i.e., DC3 sampled continental cirrus with closer proximity to convective activity). Thus, different cirrus origins and different aerosol loadings over land and ocean may both lead to different Ni in these campaigns. The aerosol indirect effects on Ni will be further discussed in Section 3.4."

*Page 9, Lines 274-278: it is confusing to me how temperature affects the variation among the phases.*

We agree that the previous discussion of temperature effects on Na was not a good explanation to the small variations seen in simulated phases. We revised this paragraph in Line 373: "The missing variations among different evolution phases indicate several potential biases in the model – 1) a lack of representation of the evolving role of heterogeneous and homogeneous nucleation (in contrast to the observations, which show higher Di initially from heterogeneous nucleation followed by lower Di from both heterogeneous and homogeneous nucleation); 2) insufficient growth of ice particles after ice nucleation indicated by lower IWC (in contrast to observed increasing trend of IWC as cirrus evolves); 3) insufficient new ice crystal formation in early and later growth phases (i.e., in contrast to observed increasing trend of Ni as cirrus evolves). We further discuss the first factor in the rest of this Section 3.4 and discuss the second and third factors through diagnosis of thermodynamic and dynamic conditions in Section 3.6."

We moved the discussion on the Na distribution with respect to temperature to a new location in Line 456 (shown in Figures S5 and S6 at the end of this response to reviewers): "In addition, to investigate whether $Na_{500}$ and $Na_{100}$ in the model have very different values compared with the observations, distributions of $Na_{100}$ and $Na_{500}$ with respect to temperature are shown in Figure S5 and S6 in the Supplement for comparisons with NSF and NASA flight campaigns, respectively. Compared with 100-km resolution observations, the model shows similar $Na_{100}$ (within 0.5 order of magnitude) and higher $Na_{500}$ by 0.5 – 1.5 orders of magnitude. This suggests the weaker aerosol indirect effects in the model are less likely caused by lower Na in the simulations."

*Page 9, Line 282: 'by subtracting the average values of the corresponding 1-degree temperature bin from the individual bin.', it is not clear how to deal with the temperature here.*

We clarified this calculation in Line 386: "Differing from the analyses in Figures 8 and 9, delta values were calculated for logarithmic IWC, Ni, Di, $Na_{100}$ and $Na_{500}$. That is, for each variable, we calculated the mean value of that variable in each 1-degree temperature bin, and then subtracted these mean values from each datum (at either 1-s or 430-s resolution) based on the temperature bin that the datum belongs to. In other words, these delta values remove the general trend of each variable with changing temperature (as shown in Figure 7)."

*Figures 14&15: please add color bars.*

Thank you for catching the missing color bars. We added the color bars to these figures (now they are Figures 16 and 17).

*Page 11, Line 343: please clarify what wsub represents.*

We clarify that wsub represents sub-grid vertical velocity fluctuations for the CAM6 model in Line 503: "According to Gettelman et al. (2010), $w_{sub}$ represents sub-grid vertical velocity for ice, which equals to the square root of two thirds of the Turbulent Kinetic Energy (TKE). Here TKE is defined in Park &

Bretherton (2009). This calculation of $w_{sub}$ in CAM6 follows the parameterization used in Morrison & Pinto (2005)."

*Page 11, Line 348: how the σw impact ice formation?*

We added a discussion on the impact of $\sigma_w$ in Line 505: "The variability of vertical velocity at sub-grid scale in the CAM6 model affects ice microphysical properties through both homogeneous and heterogeneous nucleation (Liu et al., 2007). For instance, the threshold RH for homogeneous ice formation in the slow growth regime is parameterized as a function of freezing temperature and w, while the Ni of the fast growth regime is parameterized as a function of temperature and w. Therefore, both the lower $\sigma_w$ values and lower ice supersaturation frequencies can contribute to the lower IWC and lower Ni in the model, especially for early and later growth phases, when significant amount of ice supersaturation and high $\sigma_w$ values (> 1 m s$^{-1}$) is seen in the observations but not shown in the simulations."

*Line 355: 'RHi distributions with respect to temperature', do you show that in the paper?*

We revised the sentence and clarified what Figures this sentence refers to: "The contributions of heterogeneous and homogeneous freezing have been inferred from two types of analyses, including RHi distributions with respect to temperature (Figures 4, 5 and 16) and aerosol indirect effects using linear regressions (Figures 10 – 13)."

We added references below based on response to reviewer 1's comments:

Gettelman, A., Liu, X., Ghan, S. J., Morrison, H., Park, S., Conley, A. J., Klein, S. A., Boyle, J., Mitchell, D. L., and Li, J.-L. F.: Global simulations of ice nucleation and ice supersaturation with an improved cloud scheme in the Community Atmosphere Model, J Geophys Res, 115, D18216, https://doi.org/10.1029/2009JD013797, 2010.

Liu, X., Penner, J. E., Ghan, S. J., and Wang, M.: Inclusion of Ice Microphysics in the NCAR Community Atmospheric Model Version 3 (CAM3), J Clim, 20, 4526–4547, https://doi.org/10.1175/JCLI4264.1, 2007.

Morrison, H. and Pinto, J. O.: Mesoscale modeling of springtime Arctic mixed-phase stratiform clouds using a new two-moment bulk microphysics scheme, Journal of Atmospheric Sciences, 62, 3683–3794, 2005.

Park, S. and Bretherton, C. S.: The University of Washington Shallow Convection and Moist Turbulence Schemes and Their Impact on Climate Simulations with the Community Atmosphere Model, J Clim, 22, 3449–3469, https://doi.org/10.1175/2008JCLI2557.1, 2009.

**Response to comments from Reviewer 2**

*Review on "Examination of Aerosol Indirect Effects during Cirrus Cloud Evolution" by Maciel et al. Aerosol effects on clouds constitute of one of the largest uncertainties in climate change projection. Particularly, aerosol effects on cirrus clouds are rarely studies but can be potentially important. This study investigates the aerosol indirect effects on cirrus clouds by analyzing in situ aircraft observations of cloud microphysical properties and aerosol number concentrations from NSF and NASA campaigns. The uniqueness of this study is that the analysis separates out five evolution phase of cirrus clouds so that the aerosol indirect effects can be more robustly examined. The observational data analysis is also used to compare with and evaluate the CAM6 model simulations of these field campaigns. Overall, the analysis is solid and the results are convincing. The manuscript is well written. I recommend the publication of this manuscript in ACP after my comments are addressed.*

*My major comment is that an important objective of this study is to evaluate the model simulations with observation data and some outstanding biases of the model simulations are thus identified. Therefore, it would be helpful to add some descriptions of the CAM6 model parameterizations of cloud microphysics and aerosols. I have also some minor comments as outlined below.*

We thank the reviewer for the helpful feedback. We have addressed each comment below and made respective changes to the revised manuscript.

*1. Abstract. L19-21. "Observations show stronger aerosol indirect effects (i.e., positive correlations between IWC, Ni and Na) in the Southern Hemisphere (SH) compared with the Northern Hemisphere (NH),.." It is somehow count-intuitive because there are not as many INPs (e.g., dust) in SH as in NH. The authors are suggested to add some more elucidations for this statement.*

This is a good point. We can see why the original description can be misleading. We revised the description, to say that there is higher "sensitivity" to the increase of Na in the SH than NH. This means that even though SH has lower INP concentrations, when exposed to the same amount of increase of Na, SH shows more significant increases of IWC and Ni than NH (when also exposed to the same amount of increase of Na). Please note that the linear fit (such as Figures 10-13) is conducted between the change of IWC (or Ni, Di) and the change of Na ($Na_{100}$ or $Na_{500}$), using delta values, and these delta values are relative to the mean values in each temperature bin.

We revised this sentence in the abstract in Line 24: "Observations show positive correlations of IWC, Ni and ice crystal mean diameter (Di) with respect to Na in both Northern and Southern Hemispheres (NH and SH), while the simulations show negative correlations in the SH. The observations also show higher increases of IWC and Ni in the SH under the same increase of Na than those shown in the NH, indicating higher sensitivity of cirrus microphysical properties to increases of Na in the SH than the NH."

*2. L50. "the number of ice crystals that are nucleated is affected by both strong diffusion and turbulence." The number of ice crystals that are nucleated is affected by cooling rate (or updrafts). Can the authors clarify what "diffusion" means here?*

We revised this sentence to be more readable in Line 57: "Through the use of analytical equations, Kärcher and Jensen (2017) discovered that cirrus cloud homogeneous freezing is spatially limited and very fleeting, and ice microphysical properties are affected by both strong turbulent diffusion and entrainment mixing. For example, turbulent diffusion can dilute and spread out ice crystals formed by homogeneous freezing and expose them to ice supersaturated air for further growth, while entrainment

mixing could enhance the evaporation of supercooled liquid droplets in warm cirrus regime under strong turbulence and reduce the amount of frozen droplets subsequently."

*3. L132. What do you mean "center point phase identifications"?*

This is a good question. We clarified this point in Line 164: "This timescale, i.e., 430 seconds, was chosen since it converts to a horizontal scale of about 100 km for a mean airspeed of 230 m s$^{-1}$ for all NSF campaigns and 229 m s$^{-1}$ for two NASA campaigns with a temperature less and equal to -40°C. The 430-s moving average uses one second (center point) along with 215 seconds before and 214 seconds after this second to calculate averaged values of IWC, Ni and Di, Na$_{500}$ and Na$_{100}$ on the logarithmic scale. These 430 seconds include both in-cloud and clear-sky conditions. For phase identification, cloud evolution phase number cannot be averaged (i.e., phases 1 and 2 cannot be averaged to phase 1.5), and the evolution phase identification relies on using high-resolution observations to capture the transitioning between ICRs and ISSRs. Thus, the phase number of this center-point second is used to represent the phase number of the 430-second segment."

*4. L136. When you use NCAR CAM6 to conduct simulations it would be needed to add some descriptions of CAM6 parameterizations of cloud microphysics and aerosols.*

We thank the reviewer for this suggestion. We added a paragraph describing the NCAR CAM6 model.

Line 176: "In the NCAR Community Earth System Model 2 (CESM2) CAM6 model, a prognostic moist turbulence scheme called the Cloud Layers Unified by Binomials (CLUBB) replaces the previous schemes for boundary layer turbulence, cloud macrophysics and shallow convection (Bogenschutz et al., 2013). An adjustment was also applied to the deep convection scheme to incorporate sensitivity to convection inhibition (Zhang and McFarlane, 1995). In addition, instead of treating hydrometeors, i.e., rain and snow, diagnostically, CAM6 includes an improved bulk two-moment cloud microphysics scheme that treats them prognostically (Gettelman and Morrison, 2015). For the simulations of aerosols and aerosol-cloud interactions the microphysics scheme is coupled with MAM4, a four-mode aerosol model that permits ice crystals to form through the heterogeneous nucleation of dust particles and homogeneous freezing of sulfate aerosols (Liu et al., 2007; Liu and Penner, 2005). Finally, for considerations of pre-existing ice the model follows Shi et al. (2015)."

Added references in this paragraph:

Gettelman, A. and Morrison, H.: Advanced two-moment bulk microphysics for global models. Part I: Off-line tests and comparison with other schemes, J. Climate, 28, 1268–1287, https://doi.org/10.1175/JCLI-D-14-00102.1, 2015

Shi, X., Liu, X., and Zhang, K.: Effects of pre-existing ice crystals on cirrus clouds and comparison between different ice nucleation parameterizations with the Community Atmosphere Model (CAM5), Atmos. Chem. Phys., 15, 1503–1520, https://doi.org/10.5194/acp-15-1503-2015, 2015

Bogenschutz, P. A., Gettelman, A., Morrison, H., Larson, V. E., Craig, C., and Schanen, D. P.: Higher-order turbulence closure and its impact on climate simulations in the community atmosphere model, J. Climate, 26, 9655–9676, https://doi.org/10.1175/JCLI-D-13-00075.1, 2013

Liu, X., Penner, J. E., Ghan, S. J., and Wang, M.: Inclusion of ice microphysics in the NCAR Community Atmospheric Model version 3 (CAM3), J. Climate, 20, 4526–4547, https://doi.org/10.1175/JCLI4264.1, 2007

Liu, X. and Penner, J. E.: Ice nucleation parameterization for global models, Meteorol. Zeitschrift, 14, 499–514, https://doi.org/10.1127/0941-2948/2005/0059, 2005

Zhang, G. J. and McFarlane, N. A.: Sensitivity of climate simulations to the parameterization of cumulus convection in the canadian climate centre general circulation model, Atmos.-Ocean, 33, 407–446, https://doi.org/10.1080/07055900.1995.9649539, 1995

*5. L136. Here you mention the NSF campaigns. Why do not you conduct the similar simulations for the NASA campaigns?*

We thank the reviewer for the suggestion. We added new Figures 12 and 13 (shown on the next page), as well as supplemental Figures S2 – S6, S9 and S10 (shown at the end of this response to reviewers) to complete the comparisons of nudged CAM6 simulations with both NASA and NSF flight campaigns. The main observational features (such as positive correlations of IWC, Ni and Di with respect to $Na_{100}$ and $Na_{500}$) and main model biases (such as lower IWC in the model) are similar between NSF and NASA campaigns. We found an interesting result that CAM6 simulations seem to represent aerosol indirect effects better for convective cirrus (represented by two NASA campaigns, NASA DC3 and SEAC4RS), while the aerosol indirect effects in the simulations are weaker for in-situ cirrus (represented by the majority of the NSF measurements). This result is also further investigated in Figures S3 and S4 by comparing observations and simulations for individual campaigns.

We initially did not use NASA campaigns for model evaluation mainly because the NSF and NASA campaigns measured cloud microphysical properties with two different probes (Fast-2DC in NSF and 2DS in NASA). These two probes have different measurement ranges and cannot be easily combined into one unified dataset. A new graduate student is currently working on a new method to combine the NSF and NASA campaigns to solve this problem.

For this work, we kept the model evaluation separate between NSF and NASA campaigns, and we clarified this point in Line 194: "In addition, by applying the methods from Eidhammer et al. (2014), we restricted the simulated ice and snow to match the size range of either the Fast-2DC probe in NSF campaigns or the 2D-S probe in NASA campaigns. The mass and number concentrations of ice and snow were then calculated by using the integrals of incomplete gamma functions, i.e., from 62.5 to 3200 µm for NSF campaigns, or 5 to 3005 µm for NASA campaigns. Because NSF and NASA campaigns have different size ranges for cloud hydrometeor measurements, we separately compare them with CAM6 simulations and do not combine these two observation datasets."

We added descriptions in Section 2.2 about usage of NASA campaigns for model evaluation.

Line 129: "The main purpose of the analysis of these NASA campaigns is to provide a contrast to the NSF data from different airborne platforms and instrumentations and different types of cirrus clouds being sampled. In fact, most of the NSF campaigns (except for NSF DC3 campaign) mostly sampled in-situ cirrus clouds, while several NASA campaigns such as NASA DC3, SEAC4RS and MACPEX sampled more convective cirrus."

Line 137: "For comparisons between NASA campaigns and CAM model simulations, two NASA campaigns – DC3 and SEAC4RS – are used for model comparisons, since ATTREX and POSIDON did not provide aerosol measurements and MACPEX had some issues with aerosol measurements."

The new Figures 12 and 13 are copied below. We added text to describe these figures in Line 437: "In Figures 12 and 13, similar directions of aerosol indirect effects on IWC, Ni and Di are seen in two NASA campaigns but with higher slope values, indicating slightly stronger aerosol indirect effects in convective cirrus than in-situ cirrus. The detailed information of linear fittings and their statistical significance is shown

in Table S4. Interestingly, CAM6 simulations for NASA DC3 and SEAC4RS campaigns show positive correlations of IWC and Ni with respect to Na (both $Na_{100}$ and $Na_{500}$), and negative correlation of Di with respect to Na. The simulations show stronger aerosol indirect effects in nucleation phase than early/later growth phase, as well as similar indirect effects between small and large aerosols. To further investigate the reason why model shows better agreement with observations for aerosol indirect effects on IWC and Ni during two NASA campaigns, we individually quantify aerosol indirect effects on cirrus clouds for each campaign, including six NSF and two NASA campaigns, as shown in Figures S3 and S4 in the supplement. The results show that simulations seem to better capture aerosol indirect effects for those flight campaigns targeting continental convective activity (e.g., NSF DC3, NASA DC3, and NASA SEAC4RS). This suggests that CAM6 simulations may represent the homogeneous freezing (which occurs more frequently with strong updrafts and fast cooling rate in convective cirrus) better than heterogeneous nucleation (which requires existence of INPs). This speculation is also consistent with the finding that simulations overestimate Ni and underestimate Di in nucleation phase as shown in Figures 8. Simulations show negative correlations between Di and Na while observations show positive correlations. This feature is mostly likely caused by the fact that simulations have limited amount of ice supersaturation for one-time nucleation events, and therefore ice crystals often compete for their growth (i.e., higher Ni associated with lower Di). But as we discussed above, observations may not have direct competition between multiple nucleation events if they do not happen at the same location and time as a cloud evolves. Further discussions on thermodynamic and dynamic conditions in the simulations are included in Section 3.6."

[Figure]

**Figure 12**. Similar to Figure 10, except for using two NASA flight campaigns to analyze linear regressions with respect to $dlog_{10}(Na_{100})$.

[Figure]

**Figure 13.** Similar to Figure 10, except for using two NASA flight campaigns to analyze linear regressions with respect to dlog$_{10}$(Na$_{500}$).

We also added new Figures S9 and S10 (shown at the end of this response) to compare thermodynamic and dynamical conditions between CAM simulations and NASA campaigns (Line 501): "Simulations compared with NASA campaigns also show smaller amounts of ice supersaturation and smaller variabilities of vertical velocity (Figures S9 and S10 in the supplement)."

In the abstract, we also modified the main finding to clarify that the weaker aerosol indirect effects are seen for in-situ cirrus, not for convective cirrus in Line 21: "Simulated aerosol indirect effects are weaker than the observations for both larger and smaller aerosols for in-situ cirrus, while the simulated aerosol indirect effects are closer to observations in convective cirrus. The results also indicate that simulations overestimate homogeneous freezing, underestimate heterogeneous nucleation, and underestimate the continuous formation and growth of ice crystals as cirrus clouds evolve."

*6. L152. How do you calculate Na100 and Na500 from modeled aerosol modes?*

We clarified this in Line 203: "Three modes of simulated aerosols – Aitken, accumulation and coarse aerosol modes are individually restricted to diameters > 500 nm and > 100 nm. These size-restricted aerosol number concentrations are then summed up to represent total number concentrations of aerosols larger than 100 nm and 500 nm (i.e., Na$_{100}$ and Na$_{500}$), respectively. The same size ranges of > 100 nm and > 500 nm are also derived from the UHSAS aerosol measurements for model comparisons."

*7. L164. How do you calculate the ICRs and ISSRs spatial scales?*

We clarified the calculation of ICR/ISSR spatial ratio in Line 146: "The length scale of each phase segment is calculated as the number of seconds of that segment multiplying the aircraft mean true airspeed inside that segment. The average aircraft true airspeed inside all observed evolution phases is ~ $230 \text{ m s}^{-1}$."

*8. L171. What is the implication of this result? "a cloud segment has the highest probability to be in the early growth phase (i.e., phase 3) for almost all the campaigns (except MACPEX)."*

We added more implications for this result in Line 232: "The result shows that the early growth phase (i.e., when ISSR and ICR intersect each other) has the most dominant spatial coverage for almost all the campaigns (except MACPEX). Since the ICR/ISSR ratio is also around unity, these two results indicate that the coexistence of ISSRs and ICRs at similar spatial scales provides a semi-steady state for cirrus evolution, possibly because new ice crystal formation in ISSRs can balance out the sedimentation of aged ice crystals and therefore cirrus clouds can persist in this condition."

*9. L189. Shall "0.35" be "0.2"?*

Thank you for catching this typo. It should be 0.2. We revised it.

*10. L266-270. "This feature indicates that as Na500 exceeds a threshold, the available INPs in the air parcel may have been depleted, and therefore no new ice nucleation can be initiated with additional larger aerosols that are not INPs." Here, the explanation is vague and not very convincing. there is no data beyond logNa500> 1.3. Na500 cannot be depleted at those high concentrations (>10 cm-3). What are the additional larger aerosols that are not INPs?*

We thank the reviewer for pointing this out. The original description was inaccurate. We revised this paragraph in Line 417: "In addition, IWC and Ni slowly increase with increasing $d\log_{10}Na_{100}$ when it is relatively low (i.e., $d\log_{10}Na_{100} < 1$), yet they significantly increase when $Na_{100}$ is $10 \text{ cm}^{-3}$ higher than the mean $Na_{100}$ value. On the other hand, IWC and Ni continuously increase with $d\log_{10}Na_{500}$ throughout the entire range. This feature indicates that at lower $Na_{100}$, aerosols may activate ice nucleation through homogeneous freezing such as by freezing sulfate aerosols, while at higher $Na_{100}$, there is a higher probability of having INPs that can initiate ice nucleation through heterogeneous nucleation and further expedite the formation of new ice crystals."

*11. L326. "...that increasing aerosol concentrations in the SH may be more effective in increasing ice nucleation compared with the NH." This statement is interesting but may need more elucidation.*

We added discussion on this point in Line 474: "The stronger aerosol indirect effects are seen from both smaller and larger aerosols, indicating that increasing the same amount of aerosol concentrations in the SH may be more effective in increasing ice nucleation compared with the NH. This suggests that even though on average the SH has lower INP concentrations than the NH, adding the same amount of INPs in the SH can have more significant impacts on cirrus microphysical properties compared with the NH. This feature also indicates that ice crystal formation in the SH may be more restricted by INP concentrations, while the NH may be more restricted by the availability of ice supersaturation."

*12. L331-332. Here the authors talk about the model bias related to cirrus microphysical properties. However, there is also large bias in aerosols (Na100 and Na500). Add model observation comparisons of aerosols.*

The model-observation comparisons of aerosols are shown in Figures S5 and S6 (copied at the end of this response to reviewers) for comparisons of model simulations against NSF and NASA campaigns, respectively. The discussion is added in Line 456: "In addition, to investigate whether $Na_{500}$ and $Na_{100}$ in

the model have very different values compared with the observations, distributions of $Na_{500}$ and $Na_{100}$ with respect to temperature are shown in Figures S5 and S6 in the Supplement for comparisons with NSF and NASA flight campaigns, respectively. Compared with 100-km resolution observations, the model shows similar $Na_{100}$ (within 0.5 order of magnitude) and higher $Na_{500}$ by $0.5 – 1.5$ orders of magnitude. This suggests the weaker aerosol indirect effects in the model are less likely caused by lower Na in the simulations."

*13. L342. Shall "40 seconds" be "1 second"?*

This is actually 40 seconds. The standard deviation of vertical velocity, $\sigma_w$, can only be calculated if there are more than three seconds. Thus, we chose to calculate $\sigma_w$ for every 40 seconds or every 430 seconds. We clarified this point: "To better compare with model results, $\sigma_w$, or standard deviation of w, is calculated for every 40 seconds or 430 seconds of observations at 1-Hz resolution."

*14. L348. "Both the lower σw values and lower ice supersaturation frequencies can contribute to the lower IWC in the model". How about modeled Ni?*

We added more discussions on what this means for Ni in Line 505: "The variability of vertical velocity at sub-grid scale in the CAM6 model affects ice microphysical properties through both homogeneous and heterogeneous nucleation (Liu et al., 2007). For instance, the threshold RH for homogeneous ice formation in the slow growth regime is parameterized as a function of freezing temperature and w, while the Ni of the fast growth regime is parameterized as a function of temperature and w. Therefore, both the lower $\sigma_w$ values and lower ice supersaturation frequencies can contribute to the lower IWC and lower Ni in the model, especially for early and later growth phases, when significant amount of ice supersaturation and high $\sigma_w$ values ($> 1$ m s$^{-1}$) are seen in the observations but not shown in the simulations."

*15. L378. Here the authors talk about the model sub-grid scale supersaturation and w and their impact on IWC. Why is the modeled Ni relatively better simulated than IWC?*

This is a good question. We added more discussion on the evaluation of simulated IWC versus simulated Ni, since the results have some interesting implications.

When describing Figure 7 (IWC, Ni and Di in relation to temperature), we wrote in Line 335: "The lower IWC, higher Ni and lower Di in relation to temperature seen in the model have also been shown in Patnaude et al. (2021)." But then we clarified that this does not mean that the model always shows higher Ni in all phases. We added the discussion in Line 369: "Differing from Figure 7 which shows that model overestimates Ni at various temperatures, Figures 8 and 9 show that model initially overestimates Ni for nucleation phase, but then underestimates Ni for later growth phase, indicating that the model initially overestimates homogeneous freezing but later has insufficient formation of new ice crystals as cirrus evolves."

From a first look, it may seem that the model shows more similar Ni to the observations, while the simulated IWC is a lot worse. But with a closer inspection, we notice that the simulated Ni is initially overestimated probably due to too much contribution from homogeneous freezing, but later Ni is underestimated probably due to insufficient new ice particle formation. Thus when looking at the averaged Ni with respect to temperature, the overestimation of earlier phases and underestimation of later phases cancel out each other and show similar Ni compared with observations.

For the IWC, at earlier phases, Ni is overestimated, and Di is underestimated due to too much homogeneous freezing, thus IWC in phase 2 is close to the observed IWC. But as cirrus evolves, Ni becomes underestimated, and Di is still underestimated, therefore showing a larger bias in IWC in later

phases. Both simulated IWC and Ni show weak aerosol indirect effects for in-situ cirrus as well as smaller variations among different cirrus evolution phases in both in-situ and convective cirrus.

We re-organized our summary paragraph on the main model biases to reflect new discussion on Ni in Line 531: "Several main model limitations have been identified, including lower IWC and lower Di for all phases, higher Ni for nucleation phase and lower Ni for later growth phase, and lack of variations in ice microphysical properties as cirrus clouds evolve. For the underestimation of IWC and Di in the model, Patnaude et al. (2021) has previously identified part of this issue but did not provide an investigation on the possible causes. In this work, after separating five evolution phases, we categorize the potential problems in the model simulations of cirrus clouds into three main areas, including (1) overestimation of homogeneous freezing and underestimation of heterogeneous nucleation in the nucleation phase, (2) insufficient ice crystal formation and growth in early and later growth phases, and (3) weaker aerosol indirect effects for in-situ cirrus. First, the possible overestimation of homogeneous freezing in the CAM6 model is indicated by the overestimation of Ni and underestimation of Di in phase 2 (Figures 8 and 9). Such underestimation of Di is most severe for phase 2 compared with later phases. In fact, both NSF and NASA observations show larger aerosol indirect effects of larger aerosols in phase 2 compared with later phases, indicating that heterogeneous nucleation plays a more significant role early on (Figures 11 and 13). Second, as cirrus evolves, simulated Ni biases change from overestimation in phase 2 to underestimation in phases 3 and 4, and simulated IWC becomes more severely underestimated in phases 3 and 4, indicating insufficient ice crystal formation and growth. This feature is corroborated by the lower frequency of ice supersaturation and lower magnitudes of vertical velocity fluctuations seen in phases 3 and 4 in the simulations (Figures 16 and 17, respectively). Third, the weaker aerosol indirect effects seen in simulations for in-situ cirrus compared with convective cirrus also suggests that the model more proficiently represents homogeneous freezing than heterogeneous nucleation, since the former mechanism happens more frequently around convective activity with higher ice supersaturation."

Below are the new Figures S2 – S6, S9 and S10 from the revised supplemental information.

[Figure]

**Figure S2.** Similar to Figure 7, the geometric means of $\log_{10}(\text{IWC})$, $\log_{10}(\text{Ni})$, and Di plotted against temperature for the in-cloud evolution phases using the 1-Hz observational data, the 430-s averaged data and the model data for the two NASA campaigns in columns 1, 2, and 3, respectively. The number of samples is shown in the last row.

[Figure]

**Figure S3.** Linear regressions of IWC versus dlog₁₀(Na₁₀₀) plotted by individual campaigns for (a, c) 1-Hz observations and (b, d) model simulations. (e-h) Number of samples for a-d, respectively. Panels a, b, e and f show the evolution phases of 2, 3, or 4, while panels c, d, g and h show evolution phase 5.

[Figure]

**Figure S4.** Similar to Figure S3, except for linear regressions of IWC versus dlog$_{10}$(Na$_{500}$).

[Figure]

**Figure S5.** Averages of $\log_{10}(Na_{100})$ and $\log_{10}(Na_{500})$ in each 2-degree temperature bin for evolution phases 2 to 5 using 1-s observations, 430-s observations, and the simulations in column 1, 2 and 3, respectively. The number of samples is shown in the last two rows for $\log_{10}(Na_{100})$ and $\log_{10}(Na_{500})$. The comparisons are against the NSF dataset.

[Figure]

**Figure S6.** Similar to Figure S5, but for comparisons against two NASA campaigns (i.e., DC3 and SEAC4RS).

[Figure]

**Figure S9.** Same as Figure 16 except for comparing model simulations with two NASA campaigns.

[Figure]

**Figure S10.** Same as Figure 17 except for comparing model simulations with two NASA campaigns.